# CENTROID APPROXIMATION FOR BYZANTINE-TOLERANT FEDERATED LEARNING

## ABSTRACT

Federated learning allows each client to keep its data locally when training machine learning models in a distributed setting. Significant recent research established the requirements that the input must satisfy in order to guarantee convergence of the training loop. This line of work uses averaging as the aggregation rule for the training models. In particular, we are interested in whether federated learning is robust to Byzantine behavior, and observe and investigate a tradeoff between the average/centroid and the validity conditions from distributed computing. We show that the various validity conditions alone do not guarantee a good approximation of the average. Furthermore, we show that reaching good approximation does not give good results in experimental settings due to possible Byzantine outliers. Our main contribution is the first lower bound of $\min\{\frac{n-t}{t}, \sqrt{d}\}$ on the centroid approximation under box validity that is often considered in the literature, where $n$ is the number of clients, $t$ the upper bound on the number of Byzantine faults, and $d$ is the dimension of the machine learning model. We complement this lower bound by an upper bound of $2\min\{n, \sqrt{d}\}$, by providing a new analysis for the case $n < d$. In addition, we present a new algorithm that achieves a $\sqrt{2d}$-approximation under convex validity, which also proves that the existing lower bound in the literature is tight. We show that all presented bounds can also be achieved in the distributed peer-to-peer setting. We complement our analytical results with empirical evaluations in federated stochastic gradient descent and federated averaging settings.

## 1 INTRODUCTION

Federated learning (FL) (McMahan et al., 2016; 2017) is a decentralized technique for training machine learning models based on sharing model parameters while keeping the training data *locally*. In this work, we are particularly interested in the setting where the clients share updates — namely either the gradients in case of *federated stochastic gradient descent (FedSGD)* or the model parameters in case of *federated averaging (FedAvg)* — with a trusted central server. After the server has received the updates, it aggregates the results, updates the model parameters, and then shares the new model parameters with the clients for the next training round. This technique is popular when data privacy requirements prevent clients from sharing their data directly with the server (Zhang et al., 2021; whi, 2013; Kairouz et al., 2021). The most common aggregation rule used to select a representative vector (gradient or model parameters) is averaging (McMahan et al., 2016; 2017; Zhao et al., 2018; Reddi et al., 2021; Karimireddy et al., 2020; Mitra et al., 2021; Wang et al., 2020; Li et al., 2020; Jhunjhunwala et al., 2023; 2022). However, when averaging is used, training can fail if some clients do not behave as expected. In particular, a single faulty vector can arbitrarily shift the average in any direction, leading to erroneous updates of the model parameters. Especially in the context of federated learning, it is crucial to be robust to malicious behavior and Byzantine faults, which is also the focus of our paper. In the case of homogeneous training data, it is usually possible to use similarities between vectors to exclude such outliers (Fang et al., 2022; Yang & Bajwa, 2019; El-Mhamdi et al., 2020). If the data is heterogeneous, such similarities may not exist.

Previously proposed Byzantine-tolerant federated learning methods for heterogeneous datasets focus on showing convergence of the training process and apply statistical methods for vector aggregation (Data & Diggavi, 2021; Li et al., 2019; Ghosh et al., 2019). To mitigate Byzantine behavior,

their methods remove outliers from the data and make additional assumptions on the input vectors of the clients. An alternative approach is to use the absolute or average distance to the average to evaluate federated learning algorithms (El-Mhamdi et al., 2021). This absolute measure, however, only allows one to analyze the worst-case Byzantine attack. Another measure that incorporates Byzantine vectors is $(f, \kappa)$-robustness Allouah et al. (2023). In Appendix A, we show that this robustness measure misclassifies optimal solutions under Byzantine failures. Recently, a new approximation measure was introduced to estimate the quality of an aggregated average in a Byzantine environment (Cambus & Melnyk, 2023) for approximate agreement algorithms. This approximation measure allows one to not only analyze the worst-case input setting, but rather estimate the quality of an algorithm based on the given input distribution.

In this work, we transfer the idea of approximating the average vector to the traditional federated learning setting with $n$ clients and one trusted server. In distributed computing, validity conditions are used to restrict an algorithm from terminating on arbitrary inputs. We investigate the trade-off between the validity conditions and the approximation of the average vector for federated learning. This allows us to present aggregation algorithms that perform well under different input distributions.

**The benefits of average approximation.** We consider the approximation of the average to evaluate the quality of our algorithms. As we motivate in the following, a low average approximation ratio implies that an algorithm performs well for a given input distribution. Formally, given $n$ vectors, up to $t$ of which can be Byzantine, an optimal choice of the average vector under Byzantine attacks is defined as the midpoint of the smallest ball $B$ that encloses each average obtained from every subset of $n - t$ vectors. When $t$ clients are Byzantine, exactly one of these averages was computed from only non-faulty vectors. Therefore, the midpoint minimizes the maximum distance to the non-faulty average vector in the worst case. The approximation ratio is then defined as the ratio between the distance from the aggregation vector to the non-faulty average, and the radius of $B$.

The main advantage of this approximation ratio is that it is defined relative to the input setting: In scenarios with heterogeneous training data, Byzantine vectors cannot be differentiated from non-faulty vectors. That is, a large radius of the minimum covering ball either represents "bad" Byzantine behavior, or a "bad" initial configuration where each client has vastly different input. In such a scenario, no aggregation algorithm can choose a representative average vector. The large ball radius prevents one from punishing an algorithm for a large absolute distance to the average vector. A small radius, on the other hand, represents "benign" Byzantine behavior and very similar inputs. In such a scenario, an aggregation algorithm should be able to choose an aggregation vector that is close to the original average. Figure 1 visualizes the continuous change in the ball radius depending on the input vectors of the clients.

**Contributions.** We first show that known validity conditions from the literature do not guarantee good approximation of the average. We then show that under weak and strong validity conditions, both of which only require the server to output the same vector as the non-faulty client if all non-faulty clients send the server the same vector, a constant approximation of the average can be achieved.

Our first main contribution is almost tight bounds for algorithms that satisfy box validity, where the aggregation vector lies in the coordinate-parallel hyperbox of non-faulty vectors. We present a lower bound of $\min\{\sqrt{(n-t)/t}, \sqrt{d}\}$ for the centroid approximation and show that the existing Box algorithm can achieve an approximation of $2\sqrt{\min\{n, d\}}$ by providing a new upper bound proof for the case $n < d$. Our second main contribution is a tight upper bound (a $2d$-approximation) for convex validity, where the aggregation vector lies in the convex hull of all vectors. Convex validity is the predominant validity condition used to solve multidimensional Byzantine agreement in the literature Mendes et al. (2015); Ghinea (2025). Note that this setting is only of theoretical interest to this work, as it requires the number of clients to be larger than the dimension of their input vectors ($n > (d+1) \cdot t$). In fact, Xiang & Vaidya (2017) show that box validity is the only $k$-relaxed convex hull, with $k = 1$, that allows reducing the number of nodes from $n > (d+1) \cdot t$ to $n > 3t$. We show that all presented bounds can also be achieved in the distributed peer-to-peer setting. The agreement algorithms presented differ from (El-Mhamdi et al., 2021; Cambus & Melnyk, 2023), since only exact agreement is considered in this paper.

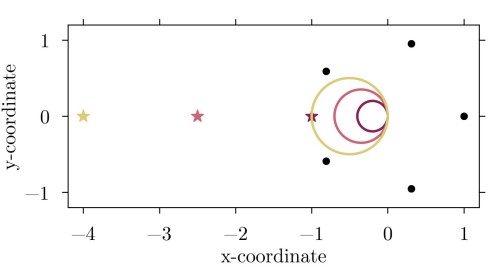
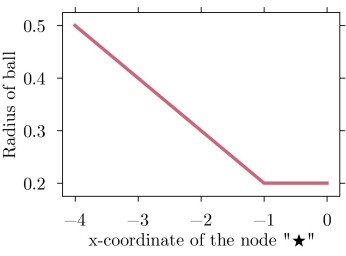

(a) Distribution of the input vectors    (b) Radius of the ball of averages

Figure 1: This figure shows how the radius of the smallest ball containing all averages depends on different distributions of the inputs. There are 6 clients, one of which is possibly Byzantine. On the left, the points represent fixed input vectors. The three stars represent three different scenarios of the input of the sixth client. The circles represent the smallest balls containing all possible averages on subsets of five points. On the right, the radius of the minimum covering ball is presented when the $x$-coordinate of the last client is moved from $-4$ to $0$. Observe that the radius of the minimum covering ball cannot be zero, as any of the points in the figure are also potentially Byzantine. The yellow scenario is a "bad" input setting where there is either one non-faulty client with very distinct data, or a Byzantine party tries to disrupt the training process. The dark red scenario is a benign setting where an aggregation algorithm should be able to output a vector close to the actual centroid. Accordingly, the radius of the ball is small in this scenario.

Finally, we extend our analytical results with simulations. Since the dimension of the data is usually much larger than the number of clients in federated learning, we focus on algorithms that satisfy box validity and weaker validity conditions. In our evaluation, we differentiate between the settings where the gradients (*FedSGD*) and the model parameters (*FedAvg*) are aggregated, and show how selected algorithms perform under different failure scenarios.

## 1.1 RELATED WORK

Dean et al. (2012) proposed a first distributed solution to train a large machine learning model on tens of thousands of CPU cores. Their work initiated a study of asynchronous algorithms for distributed stochastic gradient descent (SGD) that focus on scalability and communication efficiency (Li et al., 2014a;b; Zhang et al., 2013; Shamir et al., 2014). The synchronous version of SGD has been proposed by Chen et al. (2016). We refer to this framework as *FedSGD*. *FedSGD* has also been considered under Byzantine adversaries, both in synchronous (Alistarh et al., 2018; El Mhamdi et al., 2018) and asynchronous settings (Damaskinos et al., 2018). While the mentioned work assumes homogeneous data distributions, some efforts have also been made to incorporate data heterogeneity (Li et al., 2019; Xie et al., 2019; Ghosh et al., 2019; Data & Diggavi, 2021). To tackle Byzantine behavior of the clients, these approaches make use of homogeneity of the data, apply statistical methods, or try to detect Byzantine behavior.

Federated averaging was introduced by McMahan et al. (2016; 2017) to perform training where the data is private, unbalanced, non-IID, and distributed across mobile devices. Here, model parameters instead of gradients are exchanged with a server. We refer to this framework as *FedAvg* in this paper. Much of the follow-up work has focused on showing convergence of the models in this framework without failures (Mitra et al., 2021; 504, 2021; Wang et al., 2020; Jhunjhunwala et al., 2023; 2022; Jee Cho et al., 2022). Byzantine-tolerant approaches have been introduced also for this setting, where the goal is to remove Byzantine behavior via stochastic quantization and outlier detection mechanisms (So et al., 2021).

In contrast to previous work, we do not focus on removing Byzantine clients from the training process, as such a process may influence the accuracy when the data is heterogeneous and no malicious behavior is present in the system. Instead, we use the approximation definition for the average from Cambus & Melnyk (2023) that naturally incorporates Byzantine clients. In contrast to (Cambus & Melnyk, 2023), we consider a stronger model without agreement, which makes our lower bound results more powerful, and introduce new algorithms that achieve an optimal approximation.

## 2 MODEL AND DEFINITIONS

We consider a client/server setting with one server and $n$ clients. The goal is to train a global neural network on the server with data spread heterogeneously among clients. In order to train the global model without gathering data from clients, each client possesses its own copy of the model and then shares only vectors generated from their local data and model with the server. The server then needs to aggregate the received vectors to advance the training of the global model. The training process is performed in synchronous rounds.

On top of the training set-up, we consider that up to $t < n/2$ of the clients can be Byzantine, i.e., they can behave arbitrarily, can collude, and are not bound to following the protocol. The aggregation algorithms used by the server hence need to account for this. Note that we use the standard assumption from distributed computing that Byzantine clients are not differentiable from non-faulty clients as long as they follow the protocol and only lie about their input. We treat all clients equally and do not weigh their inputs based on the size of their local dataset. This is because we derive bounds based on the number of Byzantine clients as opposed to bounds based on the size of the clients' local datasets.

The focus of this work is on the aggregation function. Consider a specific communication round, in which each client sends a vector to the server, and the server aggregates those vectors. To account for the potential presence of Byzantine clients in the system, the aggregation algorithm used by the server needs to compute an aggregation that is as little influenced by Byzantine vectors as possible. In this work, we focus on the most common aggregation rule in federated learning – the averaging aggregation rule. Since Byzantine clients can be present in the system and are undetectable, an aggregation algorithm cannot determine the centroid of vectors of non-faulty clients. We are therefore interested in the quality of the computed aggregated vector.

**Centroid approximation.** Let $\{v_i, i \in [n]\}$ be the set of all input vectors (gradients or model parameters) that each client starts with at the beginning of an aggregation step. We refer to this set as the input layout. Note that up to $t < n/2$ of those vectors could be faulty or not sent because of Byzantine behavior. We assume that the server receives up to $m$ vectors $\{v_i, i \in [m]\}$, where $n-t \leq m \leq n$. Each vector $v$ is in the normed vector space $(\mathbb{R}^d, \|\cdot\|_2)$, where $\forall x = (x_1, \ldots, x_d) \in \mathbb{R}^d, \|x\|_2 = \sqrt{\sum_{k=1}^{d} x_k^2}$ and the distance between any two vectors $v$ and $w$ is their Euclidean distance $\text{dist}(v, w) = \|v - w\|_2$. When not specified, $\|\cdot\|$ refers to the 2-norm. We use the following definition of the average/centroid:

**Definition 2.1** (Centroid). *The* centroid *of a finite set of $k$ vectors $\{v_i, i \in [k]\}$ is $\frac{1}{k}\sum_{i=1}^{k} v_i$.*

We define the centroid approximation as in (Cambus & Melnyk, 2023). Let $\text{Cent}^\star$ be the centroid computed from non-faulty vectors only. Note that there can be up to $n$ non-faulty vectors as $t$ is only an upper bound on the number of Byzantine clients. In the following, we define the set of candidate centroids, which are computed based on the worst case where exactly $t$ vectors are Byzantine.

**Definition 2.2** (Set of candidate centroids). *Let $L = \{v_i, i \in [n]\}$ be the input layout. The* set of candidate centroids*, denoted $\text{S}_{\text{Cent}}$, is defined as*

$$\text{S}_{\text{Cent}} := \left\{ \frac{1}{n-t}\sum\nolimits_{i \in I} v_i \Big| \forall I \subseteq [n] \; s.t. \; |I| = n-t \right\}.$$

Observe that $\text{S}_{\text{Cent}}$ cannot always be computed by an algorithm, if the algorithm does not receive all the input vectors. Instead, the algorithms compute a subset of $\text{S}_{\text{Cent}}$. Since Byzantine clients are not differentiable from non-faulty clients as long as they follow the protocol, we can only define the centroid approximation based on the worst case where exactly $t$ clients are Byzantine. We define the point minimizing the maximum distance to all vectors in the set of candidate centroids defined above as the center of the following ball:

**Definition 2.3** (Minimum covering ball Elzinga & Hearn (1972)). *The minimum covering ball* $\text{Ball}_{\text{cov}}(\text{S}_{\text{Cent}})$ *is the smallest ball containing all vectors in* $\text{S}_{\text{Cent}}$*. Its radius is denoted* $\text{Rad}_{\text{cov}}$*.*

Finally, the centroid approximation is defined as follows:

**Definition 2.4** (Centroid approximation). *Let $f \leq t$ the actual number of Byzantine faults. Given an input layout $L = \{v_i, i \in [n]\}$, let $O_{\mathcal{A}}$ be the output of an algorithm $\mathcal{A}$ computing an approximation*

*of the centroid of the $n - f$ non-faulty vectors. The approximation ratio of $\mathcal{A}$ given $L$ is the smallest $\alpha$ s.t. $\mathrm{dist}(O_\mathcal{A}, \mathrm{Cent}^\star) \leq \alpha \cdot \mathrm{Rad}_{\mathrm{cov}}$. The algorithm $\mathcal{A}$ is said to compute an $\alpha$-approximation of the centroid if, for all input layouts $L$, the approximation ratio of $\mathcal{A}$ given $L$ is upper bounded by $\alpha$.*

To satisfy box validity, we will define algorithms that rely on a less restrictive area than $\mathrm{Ball}_{\mathrm{cov}}(\mathrm{S}_{\mathrm{Cent}})$:

**Definition 2.5** (Centroid hyperbox)**.** *The* centroid hyperbox *CH is the smallest coordinate-parallel hyperbox containing* $\mathrm{S}_{\mathrm{Cent}}$.

**Validity conditions.** We noted above that a Byzantine client can shift the centroid of vectors of all clients arbitrarily, and thus it can also shift the midpoint of the minimum covering ball arbitrarily far away from $\mathrm{Cent}^\star$. Just choosing the center as the centroid approximation might not be sufficient to ensure that we can trust the output of a certain algorithm. We hence take inspiration from the distributed agreement algorithms and use validity conditions to get additional guarantees on the output of different algorithms, complementing the guarantees given by the centroid approximation ratio. A validity condition is satisfied when the output of an algorithm is guaranteed to be in a specific area, depending only on the input layout. In this work, we focus on common validity conditions from the literature. The following notation is used for one of the validity definitions:

**Notation 1.** *The smallest coordinate-parallel hyperbox containing only non-faulty vectors is called the trusted hyperbox and denoted TH.*

**Definition 2.6** (Validity conditions)**.** *Let $n$ denote the number of clients, up to $t$ of which can be Byzantine. An algorithm $\mathcal{A}$ satisfies*

**weak validity** *(Civit et al., 2022; 2021; Yin et al., 2019) if, when all clients are non-faulty and all input vectors $v_i$ are equal to a single vector $v$, the output of $\mathcal{A}$ is $v$;*

**strong validity** *(Bar-Noy & Dolev, 1988; Bracha, 1987; Bracha & Toueg, 1983) if, when all non-faulty input vectors $v_i$ are equal to a single vector $v$, the output of $\mathcal{A}$ is $v$;*

**box validity** *(Cambus & Melnyk, 2023; Dolev et al., 1986; Melnyk & Wattenhofer, 2018) if the output of $\mathcal{A}$ is inside TH (see Notation 1);*

**convex validity** *(Abbas et al., 2022; Mendes et al., 2015; Wang et al., 2019) if the output of $\mathcal{A}$ is inside the convex hull of all non-faulty input vectors.*

Note that TH cannot be computed in practice. However, we prove in Section 3 (Theorem 3.2) that an algorithm satisfies the box validity condition if and only if it agrees inside a hyperbox called the trimmed trusted hyperbox (TTH):

**Definition 2.7** (Trimmed trusted hyperbox)**.** *Let $v_1, \ldots, v_m$ be the received input vectors, where $m$ is the number of received messages. The number of Byzantine values for each coordinate is at most $m - (n - t)$. Denote $\phi : [m] \to [m]$ a bijection s.t. $v_{\phi(j_1)}[k] \leq v_{\phi(j_2)}[k], \forall j_1, j_2 \in [m]$. The trimmed trusted hyperbox is the Cartesian product of $TTH[k] := \left[ v_{\phi(m-(n-t)+1)}[k], v_{\phi(n-t)}[k] \right]$ for all $k \in [d]$.*

In a similar manner, it is proved in (Cambus & Melnyk, 2023) that, in order to satisfy the convex validity condition, an algorithm must agree inside the following area:

**Definition 2.8** (Safe area (Mendes et al., 2015))**.** *Consider $m$ vectors $\{v_1, \ldots, v_m\} =: V$, $n - t \leq m \leq n$, $t < n/(\max\{3, d+1\})$ of which can be Byzantine. Let $C_1, \ldots, C_{\binom{m}{n-t}}$ be the convex hulls of every subset of $V$ of size $n - t$. The safe area is the intersection of these convex hulls: $\bigcap_{i \in \left[ \binom{m}{n-t} \right]} C_i$.*

## 3 Centroid approximation in Byzantine-tolerant FL

In this section, we first consider approximation guarantees that are given by validity conditions only. We show that only the box validity condition guarantees a bounded approximation ratio of the $\mathrm{Cent}^\star$. In the second part, we consider the best possible approximation that can be achieved under various validity conditions. We provide tight approximation bounds for each validity condition, apart from the box validity condition, where a gap remains for some specific values of $n$ and $d$. We

conclude this section with a discussion on how our results can be transferred to federated learning in a peer-to-peer network.

### 3.1 Approximation guarantees given by validity conditions

In Appendix B.1, we show that weak, strong, and convex validity conditions are not sufficient to guarantee that an algorithm achieves a bounded approximation ratio of $\text{Cent}^\star$ (see Lemma B.1— Lemma B.3). This is because it is possible to build specific inputs for which there exists an algorithm satisfying the respective validity condition such that the output of the algorithm is at a nonzero distance from the centroid of non-faulty vectors and the minimum covering ball is reduced to a single point. Thus, satisfying the validity condition alone is not sufficient for an algorithm to be guaranteed to have a bounded approximation ratio of the centroid of non-faulty vectors. On the other hand, box validity allows one to achieve a $t/(n-t) \cdot 2 \cdot \sqrt{d}$-approximation in the worst case (Lemma B.4).

### 3.2 Upper and lower bounds for centroid approximation

In this section, we present upper and lower bounds for centroid approximation under different validity conditions. An overview of these results is presented in Table 1. Note that most bounds are tight. Only in the case $n < d$, there is a gap for approximation under box validity that remains to be investigated. Due to their simplicity or prior knowledge, the bounds for weak and strong validity are presented in the appendix (Lemma B.5—Lemma B.7). In (Cambus & Melnyk, 2023), a lower bound of $2d$ has been presented for convex validity for the worst case where $n = (d+1)t+1$. In Appendix B.2, we generalize this bound to hold for any $n > (d+1)t$ (see Lemma B.8). We next give an upper bound result for the box validity condition. Note that there are two algorithms in the literature that achieve the same approximation ratio.

| validity | LB for $n > (d+1)t$ | LB for $n < (d+1)t$ | upper bound |
|---|---|---|---|
| weak | 1 | 1 | 1 (Lemma B.5) |
| strong | 2 (Cambus & Melnyk, 2023) | 2 (Cambus & Melnyk, 2023) | 2 (Lemma B.6) |
| box | $\sqrt{d}$ (Lemma 3.3) | $\min\{\frac{n-t}{t}, \sqrt{d}\}$ (Lemma 3.3) | $2\sqrt{\min\{n,d\}}$ (Lemma 3.1) |
| convex | $2d$ ((Cambus & Melnyk, 2023), Lemma B.8) | not possible (Mendes et al., 2015) | $2d$ (Lemma 3.4) |

Table 1: This is an overview of the results established in this section. Already known results are cited in the respective cells. The lower bound on weak validity follows from the definition of approximation.

**Lemma 3.1** (Upper bound for box validity). *One round of the Box algorithm (Cambus & Melnyk, 2023) or the RB-TM algorithm (El-Mhamdi et al., 2021) achieves an approximation ratio of $2\sqrt{\min\{n,d\}}$, where $n > 2t$.*

*Proof.* Note that both algorithms were presented to solve approximate agreement. We can however let the server run one round of these algorithms as if the server were one of the nodes in the distributed network. In (Cambus & Melnyk, 2023), it was shown that the output vector of one node at the end of a round is inside the intersection of CH and TTH. Note that in their analysis the two boxes are non-empty and intersect already if $n > 2t$. This condition is sufficient to achieve a $2\sqrt{d}$-approximation (Cambus & Melnyk, 2023). This solves the case $n > d$ for $n > 2t$.

We next consider the case $n < d$. Note that if CH has dimension $n$, the diagonal length argument from (Cambus & Melnyk, 2023) implies a $2\sqrt{n}$ bound on the approximation ratio. Suppose that CH has dimension $d'$ where $n < d' \le d$. Since there are $n$ input vectors and all elements of $S_{\text{Cent}}$ are computed from those vectors, $\text{Conv}(S_{\text{Cent}})$ has to be contained in a subspace $U_{\text{input}}$ of dimension $n$. The hyperbox CH of dimension $d'$ is the smallest possible hyperbox containing the convex polytope $\text{Conv}(S_{\text{Cent}})$. Hence, $\text{Conv}(S_{\text{Cent}})$ has to intersect all $2d'$ faces of CH, otherwise there exists a hyperbox strictly contained in CH that contains $\text{Conv}(S_{\text{Cent}})$. For the sake of simplicity, assume that CH is the unit hypercube of dimension $d'$ placed at the origin with non-negative coordinates only. Note that translation and rotation of all points do not influence the approximation ratio. Further, all following computations can be adjusted with the length of the longest edge of CH to achieve the same result in the general case.

Observe that $\mathrm{Conv}(\mathrm{S_{Cent}})$ has to intersect all faces of CH that contain the origin. Consider the set of centroids in $\mathrm{S_{Cent}}$ that lie on these $d'$ faces. Any two such centroids that lie on different faces are linearly independent. Since $\mathrm{Conv}(\mathrm{S_{Cent}})$ spans at most an $n$-dimensional subspace, at most $n$ centroids in this set can be linearly independent. Note that the radius of $\mathrm{Ball_{cov}}(\mathrm{S_{Cent}})$ is maximized when the centroids lie on intersections of many faces. Consider the largest subset of linearly independent centroids that intersect the $d'$ considered faces (this subset can be chosen greedily). On average, each centroid in this subset lies in the intersection of at least $d'/n$ faces. Thus, at least one of these centroids must lie in the intersection of at least $d'/n$ faces of the unit hypercube. This implies that the radius of the minimum covering ball is at least $\sqrt{d'/n}/2$ (the intersection of $k$ faces is at distance $\sqrt{k}/2$ from the center of the hyperbox).

However, since the centroid of non-faulty vectors has to be contained inside $\mathrm{Conv}(\mathrm{S_{Cent}}) \subseteq \mathrm{CH}$, the distance between the output of an algorithm agreeing inside CH and $\mathrm{Cent}^\star$ centroid is at most $\sqrt{d'}$, hence the approximation ratio is at most $2 \cdot \sqrt{n}$. Hence, the approximation ratio of the hyperbox algorithm is at most $2 \cdot \sqrt{\min\{n,d\}}$. $\qquad\square$

Before addressing the lower bound for algorithms satisfying box validity, we first prove:

**Lemma 3.2.** *An algorithm satisfying box validity has to agree inside the trimmed trusted hyperbox.*

*Proof.* We assume that $t$ Byzantine parties follow the algorithm with their own (worst-case) input vectors, thus being undetectable. Let us consider a consensus algorithm such that the output vector $v$ always satisfies box validity. For the sake of contradiction, suppose this output vector is outside the trimmed trusted hyperbox. By definition of the trimmed trusted hyperbox, there exists a coordinate $k$ for which $v[k]$ is strictly larger than $n - t$ of the input vectors at coordinate $k$. Since Byzantine clients are undetectable, these $n - t$ input vectors could be the non-faulty ones. This implies that the output vector $v$ is not in the trusted box, thus violating the box validity condition. This is a contradiction. Hence, the output vector of any algorithm satisfying the box validity condition must be in the trimmed trusted hyperbox. $\qquad\square$

**Lemma 3.3** (Lower bound for box validity)**.** *The approximation ratio of any algorithm satisfying box validity is at least $\sqrt{1/2 \cdot \min\{\lfloor (n-t)/t \rfloor, d\}}$, where $t > 0$.*

*Proof.* In order to prove the lower bound on the approximation ratio, we present a construction where the trimmed trusted hyperbox consists of just one vector.

Consider a setting where $n - t - \min\left\{\lfloor \frac{n-t}{t} \rfloor t, dt\right\}$ input vectors are at coordinate $(0, \ldots, 0)$. We further assume that $t$ vectors are at coordinate $e_k = x \cdot u_k, \forall k \in \left[\min\left\{\lfloor \frac{n-t}{t} \rfloor, d\right\}\right]$, where $u_k$ is the $k^{th}$ unit vector and $x > 0$. Suppose the $t$ Byzantine vectors choose their input vectors to be $(0, \ldots, 0)$. Then, the trimmed trusted hyperbox is $(0, \ldots, 0)$.

The centroid of non-faulty vectors is $\frac{t}{n-t} \sum_{k=1}^{\min\{\lfloor (n-t)/t \rfloor, d\}} e_k$ and the distance between the trimmed trusted hyperbox and $\mathrm{Cent}^\star$ is

$$\mathrm{dist}\big(\mathrm{Cent}^\star, (0, \ldots, 0)\big) = \sqrt{\sum_{k=1}^{\min\{\lfloor (n-t)/t \rfloor, d\}} \left(\frac{t}{n-t} \cdot x\right)^2}$$

$$= \sqrt{\min\left\{\left\lfloor \frac{n-t}{t} \right\rfloor, d\right\} \cdot \left(\frac{t}{n-t} \cdot x\right)^2} = \sqrt{\min\left\{\left\lfloor \frac{n-t}{t} \right\rfloor, d\right\}} \cdot \frac{tx}{n-t}.$$

Now the radius of the minimum covering ball is at most the largest distance between two possible centroids:

$$\mathrm{Rad_{cov}} \leq \left\| \sum_{k=2}^{\min\{\lfloor (n-t)/t \rfloor, d\}} \left(\frac{t}{n-t} \cdot e_k\right) - \sum_{k=1}^{\min\{\lfloor (n-t)/t \rfloor, d\}-1} \left(\frac{t}{n-t} \cdot e_k\right) \right\|_2 = 2 \cdot \sqrt{\left(\frac{tx}{n-t}\right)^2}.$$

Note that since $t > 0$, $\mathrm{Rad_{cov}} > 0$ holds in our construction. Hence, the approximation ratio is at least

$$\mathrm{dist}\big(\mathrm{Cent}^\star, (0, \ldots, 0)\big)/\mathrm{Rad_{cov}} \geq \sqrt{1/2 \cdot \min\{\lfloor (n-t)/t \rfloor, d\}}. \qquad\square$$

Observe that in the case $t = 0$, the proposed algorithms from Lemma 3.1 compute the true centroid and thus are optimal. We finally consider convex validity. Note that no guarantees can be given for algorithms satisfying convex validity in the case $n \leq \max\{3, d+1\}t$ since the safe area cannot be guaranteed to exist in such cases. The results presented here are therefore only of interest in applications where the number of clients surpasses the dimension of the training model.

**Lemma 3.4** (Upper bound for convex validity)**.** *Consider the algorithm that outputs a vector contained in the safe area that minimizes the distance to the center of* $\mathrm{Ball}_{\mathrm{cov}}(\mathrm{S_{Cent}})$*. This algorithm computes a* $2d$*-approximation of the centroid, when* $n > \max\{3, d+1\}t$*.*

*Proof.* Observe that the algorithm computes at most a 2-approximation of $\mathrm{Cent}^{\star}$ if the *safe area* and $\mathrm{Ball}_{\mathrm{cov}}(\mathrm{S_{Cent}})$ intersect. This is because the algorithm then chooses a vector in $\mathrm{Ball}_{\mathrm{cov}}(\mathrm{S_{Cent}})$.

Now we consider the remaining case, where the *safe area* and $\mathrm{Ball}_{\mathrm{cov}}(\mathrm{S_{Cent}})$ are disjoint. Let $x$ denote the distance between the *safe area* and $\mathrm{Ball}_{\mathrm{cov}}(\mathrm{S_{Cent}})$ and let $S$ denote the closest point of the *safe area* to $\mathrm{Ball}_{\mathrm{cov}}(\mathrm{S_{Cent}})$ and $B$ the closest point of $\mathrm{Ball}_{\mathrm{cov}}(\mathrm{S_{Cent}})$ to the *safe area*, so that the distance between $S$ and $B$ is $x$. We start by projecting all input vectors orthogonally onto $\overline{S, B}$. The approximation ratio of the algorithm is computed as $(x + \mathrm{Rad}_{\mathrm{cov}})/\mathrm{Rad}_{\mathrm{cov}}$. Observe that the distance between any two centroids after their orthogonal projection onto $\overline{S, B}$ cannot increase due to the triangle inequality, while the distance between $S$ and $B$ remains unchanged. Therefore, the distance between any two projected centroids onto $\overline{S, B}$ is a lower bound on the diameter of $\mathrm{Ball}_{\mathrm{cov}}(\mathrm{S_{Cent}})$. To simplify the discussion on distances, we assume that $S$ is at coordinate 0 and $B$ is at coordinate $x$.

In the following, we will lower bound the size of $\mathrm{Rad}_{\mathrm{cov}}$ and upper bound the size of $x$. Let the projection of the vectors $v_1, \ldots, v_n$ be denoted $p_1, \ldots, p_n$ such that the vector projected on the smallest coordinate is denoted $p_1$ and the one projected on the largest coordinate is $p_n$. Note that there must be at least $t + 1$ vectors $p_i$ having negative coordinates, otherwise there would exist a convex hull of $n - t$ vectors that would project onto only strictly positive coordinates, which is a contradiction. There are also at least $t + 1$ vectors $p_j$ that have coordinate at least $x$. If this was not true, there would exists a centroid with a smaller coordinate than $x$, which is a contradiction. Further, there are at most $td$ vectors with a positive coordinate (see proof of Lemma 3.5).

Let $l$ denote the number of vectors $p_i$ with negative coordinates. Let $r$ denote the number of vectors $p_i$ with a larger coordinate than $x$, and let $y_1, \ldots, y_r$ denote the coordinates of these vectors in increasing order. Further, we say that the smallest $r - t$ coordinates have an average value of $\overline{y}_{min}$ while the largest $t$ coordinates have an average of $\overline{y}_{max}$. The average of all vectors $p_i$ with coordinates between 0 and $x$ is defined to be $a$.

Observe that $x$ is upper bounded by the coordinate of any possible centroid. We choose the following centroid to upper bound $x$: the average of some $t+1$ vectors with negative coordinates, all the vectors between 0 and $x$, and the remaining smallest $r - t$ vectors with coordinates larger than $x$. This gives the following bound:

$$x \leq \frac{1}{n-t} \left( \sum_{i=1}^{r-t} y_i + a \cdot (n - r - l) \right) \leq \frac{1}{n-t}(n - t - l) \cdot \overline{y}_{min}$$

Note that we upper bounded all vectors with coordinates smaller than 0 by 0.

To lower bound the diameter of $\mathrm{Ball}_{\mathrm{cov}}(\mathrm{S_{Cent}})$, we consider the difference between its largest and smallest coordinates:

$$\mathrm{Rad}_{\mathrm{cov}} \geq \frac{1}{2(n-t)} \left( \sum_{i=t+1}^{n} p_i - \sum_{i=1}^{n-t} p_i \right) \geq \frac{1}{2(n-t)} \left( t \cdot \overline{y}_{max} - \sum_{i=1}^{t} p_i \right) \geq \frac{t}{2(n-t)} \cdot \overline{y}_{max}$$

where $\frac{1}{n-t} \sum_{i=1}^{t} p_i \leq 0$ since there are at least $t + 1$ vectors $p_i$ with negative coordinates.

The approximation ratio achieved by the algorithm can now be upper bounded by:

$$\frac{x}{\mathrm{Rad}_{\mathrm{cov}}} + 1 \leq \frac{\frac{1}{n-t}(n - t - l) \cdot \overline{y}_{min}}{\frac{t}{2(n-t)} \cdot \overline{y}_{max}} + 1 \leq \frac{2(n - t - l)}{t} + 1 \leq \frac{2dt}{t} + 1 = 2d + 1.$$

The last inequality holds because there can be at most $dt$ vectors with positive coordinates, i.e., $n - t - l \leq dt$. $\qquad\square$

**Lemma 3.5.** *Assume that the safe area is a q-dimensional convex polytope, where $1 \leq q \leq d$. Consider the q-dimensional subspace in which the safe area is defined. Let $H$ be a hyperplane that touches the safe area and divides the q-dimensional space into two subspaces. Then, there can be at most $qt$ points on the opposite side of $H$ wrt. the safe area.*

*Proof.* Consider a vertex $s_v$ of the safe area that lies at the intersection of the *safe area* with the hyperplane $H$. Note that at least one such vertex must exist since the *safe area* is a convex polytope.

Observe that exactly $q$ $(q-1)$-faces of *safe area* meet in $s_v$. Each of these faces are hyperplanes , denoted $H_1, \ldots, H_d$, and go through $s_d$, each of them defined by a face of the safe area. The *safe area* is defined such that, for each face $F_i$, at most $t$ vectors can lie outside of *safe area* and thus on the opposite side of $H$ w.r.t. *safe area*. In total, at most $qt$ can lie on the opposite side of $H$. And at least $n - qt > n - dt$ vectors must lie inside *safe area*. $\qquad\square$

### 3.3 FEDERATED LEARNING IN PEER-TO-PEER NETWORKS

The results presented in this paper also hold for federated learning in synchronous peer-to-peer networks when $n > 3t$. In the peer-to-peer setting, there is no trusted server. Instead, the clients communicate with each other in a fully-connected network by sending messages. The aggregation step by the server is replaced by an exact Byzantine agreement algorithm that makes sure that the clients agree on the same aggregation vector. The lower bounds presented in Section 3.1 and 3.2 trivially extend to this distributed setting, as they are presented for a stronger setting in which the clients do not receive different sets of vectors as it is possible in a peer-to-peer setting. On the other hand, interactive consistency protocols (Pease et al., 1980; Fischer & Lynch, 1982) from distributed computing allow the clients to agree on the same set of vectors. Thus, each client can apply the presented aggregation algorithms locally. Since the algorithms are deterministic, all clients output identical vectors after Byzantine agreement.

## 4 PRELIMINARY EMPIRICAL INSIGHTS INTO THE TRADE-OFF

We conclude our work with simulation results for the *FedSGD* protocol. The description of the experimental setup, an analysis of different Byzantine attacks, and an evaluation of the *FedAvg* method are presented in Appendix C. The experiments are run with 100 clients under mild ($\alpha = 1$), moderate ($\alpha = 0.5$), strong ($\alpha = 0.2$) and extreme heterogeneity. Figure 2 shows how the MDA and the Box algorithms perform under the *fall of empires* attack Xie et al. (2020) with $f \in \{10, 20, 33\}$ Byzantine clients. Overall, the higher number of Byzantine clients affects the system and prevents convergence in more heterogeneous cases. For $f = 33$, MDA fails to converge across all distributions. However, the Box algorithm converges under mild heterogeneity and achieves 69% accuracy. In moderate heterogeneity case, it seems as if Box algorithm could converge, but slowly and requiring a significant amount of rounds. This implies that algorithms satisfying stronger validity conditions are more robust against Byzantine clients.

Figure 3 illustrates Fall of Empires (FoE), A Little Is Enough (ALIE), Sign Flip (SF) and mimic attack in a setting with mild heterogeneity and $f = 49$. Even though MNIST is considered a smaller dataset, the Byzantine-tolerant algorithms struggle to deal with a large numbers of Byzantine nodes. Only Box and MDA algorithm under the mimic attack converge achieving 70% and 76% accuracy, respectively. In Figure 4 we investigate the mimic attack in more detail, by considering different data distributions. Generally, having a more heterogeneous setting lowers the overall accuracy. MDA achieves higher accuracy than the Box algorithm, reflecting to the better approximation results from Table 1. However, MDA also shows small fluctuations in accuracy over rounds, as it satisfies weaker validity conditions. On the other side, box algorithm, which satisfies the box validity condition, seems very stable, but reaches slightly lower accuracy, as it provides a larger approximation of the centroid. Our experiments suggest that stronger validity conditions yield more robust solutions and that tighter centroid approximations improve accuracy. We can conclude that there is a trade-off between centroid approximation and different validity conditions.

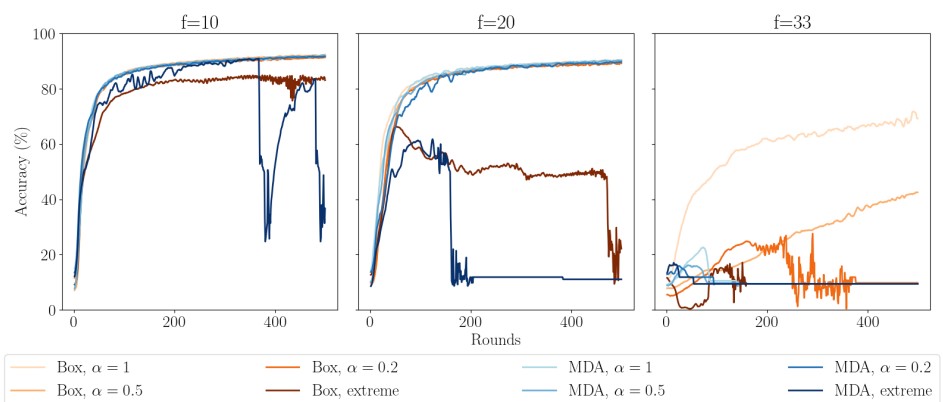

Figure 2: Fall of empires attack with $n = 100, f = \{10, 20, 33\}$ in *FedSGD* setting under mild ($\alpha = 1$), moderate ($\alpha = 0.5$), strong ($\alpha = 0.2$) and extreme heterogeneity

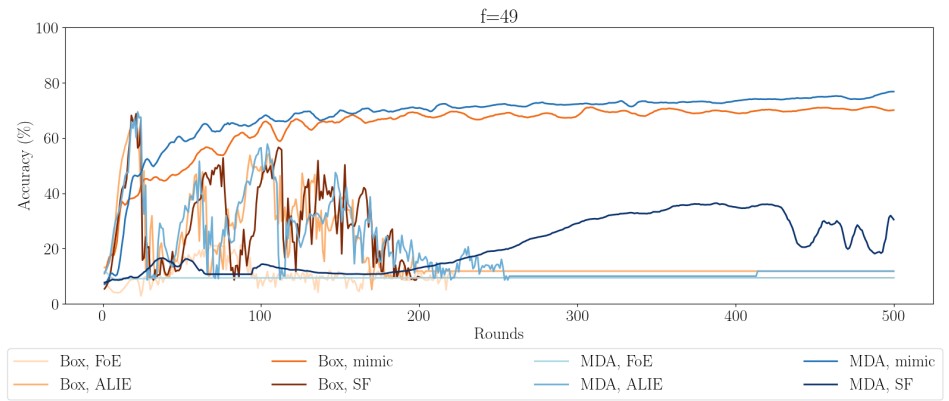

Figure 3: Attacks with $n = 100, f = 49$ in *FedSGD* setting under mild ($\alpha = 1$) heterogeneity

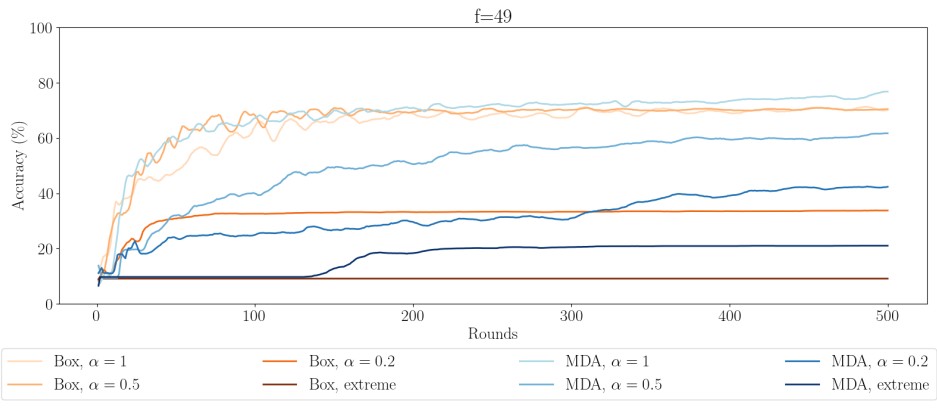

Figure 4: Mimic attack with $n = 100, f = 49$ in *FedSGD* setting under mild ($\alpha = 1$), moderate ($\alpha = 0.5$), strong ($\alpha = 0.2$) and extreme heterogeneity

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

## A    DISCUSSION OF $(f, \kappa)$-ROBUSTNESS

In this section, we review the $(f, \kappa)$-robustness definition from (Allouah et al., 2023):

**Definition A.1** ($(f, \kappa)$-robustness (Allouah et al., 2023)). *Let $f \leq t < n/2$ be the number of Byzantine nodes in the system and $\kappa \geq 0$. An aggregation rule $F$ is said to be $(f, \kappa)$-robust if for any vectors $x_1, \ldots, x_n \in \mathbb{R}^d$, and any set $S \subseteq [n]$ of site $n - f$,*

$$\|F(x_1, \ldots, x_n) - \bar{x}_S\|^2 \leq \frac{\kappa}{|S|} \sum_{i \in S} \|x_i - \bar{x}_S\|^2$$

*where $\bar{x}_S = \frac{1}{|S|} \sum_{i \in S} x_i$. $\kappa$ is here the robustness coefficient.*

Observe that for the special case where $S$ consists only of non-faulty nodes, this robustness definition is similar to the approximation definition in our paper. However, since the robustness considers every subset $S$ of $n - f$ nodes, certain Byzantine attacks can lead to bad robustness guarantees, even for an optimal algorithm:

**Example A.2.** *Consider an algorithm that outputs $\mathrm{Cent}^\star$—the centroid of non-faulty vectors. This algorithm is optimal. For simplicity, assume a setting where all input vectors lie on a line. Let the $n - 2f$ non-faulty vectors have the input value $0$, the $f$ non-faulty nodes have the input value $a > n - f$, and the $f$ Byzantine nodes have the input value $-\varepsilon$, where $\varepsilon > 0$ is an arbitrarily small constant. Observe that in the definition of $(f, \kappa)$-robustness, one subset will contain all Byzantine vectors and the $n - 2f$ non-faulty nodes with input value $0$. We denote this subset $S_{Byz}$. Also observe that the average of the non-faulty nodes is located at $\mathrm{Cent}^\star = \frac{af}{n-f}$, that the average of the nodes in $S_{Byz}$ is located at $\bar{x}_{S_{Byz}} = -\frac{\varepsilon f}{n-f}$, and that the average distance from the nodes in $S_{Byz}$ to $\bar{x}_{S_{Byz}}$ is less than $2f\varepsilon$. Therefore, $\kappa$ has to be chosen such that $\frac{af}{n-f} + \frac{\varepsilon f}{n-f} \leq \kappa \cdot 2f\varepsilon$ since $F(x_1, \ldots, x_n) = \mathrm{Cent}^\star$. We thus have the result that $\kappa \geq \frac{a}{2\varepsilon(n-f)} > \frac{1}{2\varepsilon}$. This shows that the constant $\kappa$ can however be arbitrarily large for an optimal algorithm.*

The $(f, \kappa)$-robustness provides a reasonable robustness measure when the failing nodes are outliers. However, as the above example shows, the measure is not guaranteed to be small when the failing nodes show completely arbitrary, i.e. Byzantine, behavior. Therefore, the $(f, \kappa)$-robustness definition is not suited to evaluate the quality of a Byzantine-tolerant algorithm.

## B DETAILED THEORETICAL BOUNDS

### B.1 GUARANTEES GIVEN BY THE VALIDITY CONDITIONS

**Lemma B.1.** *Satisfying weak validity is not a sufficient condition for an algorithm to achieve a bounded approximation ratio of* $\mathrm{Cent}^\star$.

*Proof.* Without loss of generality, we can consider an algorithm that either agrees on the unique input vector, or outputs the origin. Now consider the case where all clients have input $x \cdot (1, \ldots, 1)$. Then, the diameter of the minimum covering ball can be arbitrarily small, but the distance between the origin and $x \cdot (1, \ldots, 1)$ is $\sqrt{d} \cdot x$. Hence, the ratio between this distance and the radius of the minimum covering ball is unbounded.

$\square$

**Lemma B.2.** *Satisfying strong validity is not a sufficient condition for an algorithm to achieve a bounded approximation ratio of* $\mathrm{Cent}^\star$.

*Proof.* As before, we can consider an algorithm that either agrees on the unique non-faulty input vector, or outputs the origin (we do not need to know how the algorithm achieves this, only that it is a general algorithm satisfying strong validity). Assume the case, where the $n - t$ non-faulty input vectors are all $\epsilon$ away from $(1, \ldots, 1)$, and the Byzantine clients do not send any vector. The distance between the origin and the average of the non-faulty vectors is $\sqrt{d} \cdot x$. The radius of the minimum covering ball is however $0$. Hence, the approximation ratio is unbounded. $\square$

**Lemma B.3** (from (Cambus & Melnyk (2023), Observation 4.1))**.** *The worst-case approximation ratio that can be achieved by any algorithm satisfying convex validity is unbounded.*

Next, we show that the box validity condition is the only validity condition that, by itself, guarantees that any algorithm satisfying it has a bounded approximation ratio. More precisely, we show that outputting a vector inside TH is sufficient to ensure that the output is a bounded approximation of $\mathrm{Cent}^\star$.

**Lemma B.4.** *The worst-case approximation ratio that can be achieved by any algorithm satisfying box validity is at most* $\frac{t}{n-t} \cdot 2\sqrt{d}$.

*Proof.* Consider the coordinate $k \in [d]$ in which TTH realizes its longest edge. We define a bijection $\phi : [n] \to [n]$ such that, $i < j \Rightarrow v_{\phi(i)}[k] < v_{\phi(j)}[k], \forall i, j \in [n]$. Then,

$$|\mathrm{CH}[k]| = \frac{1}{n-t} \sum_{i=t+1}^{n} v_{\phi(i)} - \frac{1}{n-t} \sum_{i=1}^{n-t} v_i[k]$$

$$= \frac{1}{n-t} \sum_{i=n-t+1}^{n} v_{\phi(i)} + \frac{1}{n-t} \sum_{i=t+1}^{n-t} v_{\phi(i)} - \frac{1}{n-t} \sum_{i=1}^{t} v_i[k] - \frac{1}{n-t} \sum_{i=t+1}^{n-t} v_{\phi(i)}$$

$$= \frac{1}{n-t} \sum_{i=n-t+1}^{n} v_{\phi(i)} - \frac{1}{n-t} \sum_{i=1}^{t} v_i[k] \geq \frac{t}{n-t} v_{\phi(n-t)} - \frac{t}{n-t} v_{\phi(t)} = \frac{t}{n-t} |\mathrm{TTH}[k]|.$$

Since CH and TTH are necessarily intersecting (Cambus & Melnyk, 2023), the furthest a vector satisfying box validity can be from $\mathrm{Cent}^\star$ is if $\mathrm{Cent}^\star$ is in CH and the vector is on the opposite vertex of TTH. We showed above that the diagonal of TTH is at most $\frac{t}{n-t}$ times the diagonal of CH.

The diagonal of CH being upper bounded by $2\sqrt{d} \cdot \mathrm{Rad}_{\mathrm{cov}}$, the furthest we can be from $\mathrm{Cent}^\star$ by satisfying box validity is

$$\left(1 + \frac{t}{n-t}\right) \cdot 2\sqrt{d} \cdot \mathrm{Rad}_{\mathrm{cov}}.$$

The centroid approximation ratio of any algorithm satisfying box validity will hence be upper bounded by $\left(1 + \frac{t}{n-t}\right) \cdot 2\sqrt{d}$.

$\square$

### B.2 LOWER AND UPPER BOUNDS

In the following, we present the upper bound for weak validity.

**Lemma B.5** (upper bound for weak validity). *The best approximation ratio that can be achieved by an algorithm satisfying weak validity is 1 in the worst case.*

*Proof.* We can achieve 1 with the optimum algorithm picking the center of the minimum covering ball (see Cambus & Melnyk (2023)). This algorithm satisfies weak validity. $\square$

Note that this upper bound is tight, as the lower bound cannot be less than 1 by definition. We now present the algorithm that highlights the upper bound for strong validity.

**Lemma B.6** (Upper bound for strong validity). *The MDA algorithm (El-Mhamdi et al., 2021) outputs the average of the subset of $n - t$ vectors that have the smallest diameter, this diameter is defined as the maximum distance between any two vectors. The MDA computes a 2-approximation of the centroid, where $n > 2t$.*

*Proof.* Observe that the output vector of the MDA algorithm is in $S_{\text{Cent}}$ and is thus inside $\text{Ball}_{\text{cov}}(S_{\text{Cent}})$. The largest distance between any two vectors in $\text{Ball}_{\text{cov}}(S_{\text{Cent}})$ is upper bounded by the diameter of the ball. Thus, the algorithm computes at most a 2-approximation. $\square$

The following lemma gives a lower bound of 2 on the approximation ratio of the centroid in the context of strong validity, which matches the upper bound above. This shows that the approximation ratio of the MDA algorithm is tight.

**Lemma B.7** (Lower bound for strong validity (Cambus & Melnyk, 2023)). *The best approximation ratio that can be achieved by an algorithm satisfying strong validity is 2 in the worst case.*

We finally present the lower bound for convex validity below.

**Lemma B.8** (Lower bound for convex validity). *The best approximation ratio that can be achieved by an algorithm satisfying convex validity is at least $2d$.*

*Proof.* In (Cambus & Melnyk, 2023), a lower bound of $2d$ has been shown for the worst case $n = (d + 1)t + 1$. This proof can be easily extended to hold for the general case $n > \max\{3, d + 1\} \cdot t$. Assume that $dt$ vectors are placed at coordinates $x + \varepsilon \cdot u_i, i \in \{1, \ldots, d\}$, where $\varepsilon$ is a small constant and $t$ vectors placed at each coordinate. The remaining $n - dt$ vectors are placed at $(0, \ldots, 0)$. Assume that these $n - dt$ vectors include $t$ Byzantine vectors. Observe that such a construction is always possible since $n > (d + 1)t$.

In (Cambus & Melnyk, 2023), it was shown that the safe area of such a construction results in a single point $(0, \ldots, 0)$. Note that the non-faulty centroid is located in $td/(n - t)$, and the radius of the centroid ball is $t/(2(n - t))$. Thus, the approximation of the centroid is $2d$ in this example. $\square$

## C EMPIRICAL EVALUATION

In the practical evaluation, we differentiate between the two federated learning variants where the model parameters or the gradients are exchanged. We consider $n$ clients, where each client $i \in [n]$ has access to its own data that follows an unknown distribution $\mathcal{D}_i$. Let $F_i(x)$ be the local loss function of client $i$ with respect to model parameter $x$. The objective is

$$\arg\min_{x \in \mathbb{R}^d} F(x), \quad \text{where} \quad F(x) = \frac{1}{n} \sum_{i=1}^{n} F_i(x)$$

The training is executed in rounds. We differentiate between the following two settings:

***FedSGD*** In each round $r$, a client locally computes the gradient $g_i(x_r) = \nabla F_i(x_r)$ on its dataset. It then sends $g_i(x_r)$ to the server. The server upgrades the global model by aggregating the gradients $x_{r+1} \leftarrow x_r - \eta \frac{1}{n} \sum_{i=1}^{n} g_i(x_r)$, where $\eta$ is a fixed learning rate, and sends the new model to the clients for the next round.

***FedAvg*** In each round $r$, a client locally updates its model parameters (possibly multiple times) $x_{r+1}^i \leftarrow x_r^i - \eta g_i(x_r)$. It then shares its model parameter $x_{r+1}^i$ with the server. The server aggregates the model parameters $x_{r+1} \leftarrow \sum_{i=1}^{n} x_{r+1}^i$ and shares the new model with the clients.

The aggregation algorithm in the definition of *FedSGD* and *FedAvg* is an unweighted average of the vectors. For the experiments, we replace this aggregation step with one of the aggregation algorithms presented in Section 3.2. These aggregation algorithms are summarized below.

**Aggregation algorithms** We implemented the following aggregation algorithms for comparison:

- **Center of** $\mathrm{Ball}_{\mathrm{cov}}(\mathrm{S}_{\mathrm{Cent}})$**:** This algorithm computes all possible centroids on subsets of $n - t$ vectors and outputs the center of $\mathrm{Ball}_{\mathrm{cov}}(\mathrm{S}_{\mathrm{Cent}})$. The algorithm achieves a 1-approximation of the centroid and satisfies weak validity.

- **MDA (El-Mhamdi et al., 2021):** This algorithm computes a subset of $n - t$ vectors with the smallest diameter and outputs the centroid of this subset. The algorithm achieves a 2-approximation of the centroid and satisfies strong validity.

- **Box Algorithm (Cambus & Melnyk, 2023):** This algorithm computes the intersection of TTH and CH, and outputs the center of this intersection. In (Cambus & Melnyk, 2023), it was shown that such an intersection is non-empty for $n > 3t$. The algorithm achieves a $2\sqrt{d}$-approximation of the centroid and satisfies box validity.

We do not implement the algorithm based on the *safe area* (see Lemma 3.4), since this algorithm only works in scenarios where $n > (d + 1)t$.

## C.1 EXPERIMENTAL SETUP

We implement a client/server federated learning model for solving classification tasks in Python using the Tensorflow library. The models are evaluated on the MNIST dataset from Kaggle[1]. The dataset contains 42,000 images of handwritten digit in JPEG format which are labeled, and each class of the data is kept in a separate folder. We consider a setting with 30 clients and assume that a constant fraction of them are Byzantine. We use $f < n/3$ to denote the actual number of Byzantine clients present in the system. To simulate data heterogeneity in our experiments, we consider the Dirichlet distribution with parameter $\alpha$ Hsu et al. (2019), as done in Allouah et al. (2023); Farhadkhani et al. (2023); Allouah et al. (2025). Parameter $\alpha$ indicates the level of heterogeneity sampled by clients' datasets. Smaller values of $\alpha$ indicate a more heterogeneous setting, where a client likely owns data only from a very few classes. In line with Allouah et al. (2025), we consider three values for $\alpha$: $\alpha = 1$ representing mild heterogeneity, $\alpha = 0.5$ representing moderate heterogeneity, and $\alpha = 0.1$ representing strong heterogeneity. Additionally, we consider the extreme heterogeneous case, where the data is sorted by classes and distributed among clients such that each client possesses up to two different classes of data. Note that in Section 4 we considered a setting with $n = 100$ clients and strong heterogeneity with $\alpha = 0.2$. That is because the data distribution with $\alpha = 0.1$ created clients with less images than what is required to sample a batch, so training these clients is not possible.

The underlying neural network for solving the image classification task is a MultiLayer Perceptron (MLP) with 3 layers. The learning rate is set to $\eta = 0.01$ and the decay is calculated with respect to the number of global communication rounds (epochs), i.e. $decay = \frac{\eta}{rounds}$. The batch size is set to 32.

Byzantine behavior in federated learning has been extensively studied in the literature, and the attacks has been categorized into training-based and parameter-based attacks (Shi et al., 2022). Training-based attacks, also known as data poisoning attacks, have been analyzed in (Biggio et al.,

---

[1]https://www.kaggle.com/datasets/scolianni/mnistasjpg, accessed on 25.09.2025

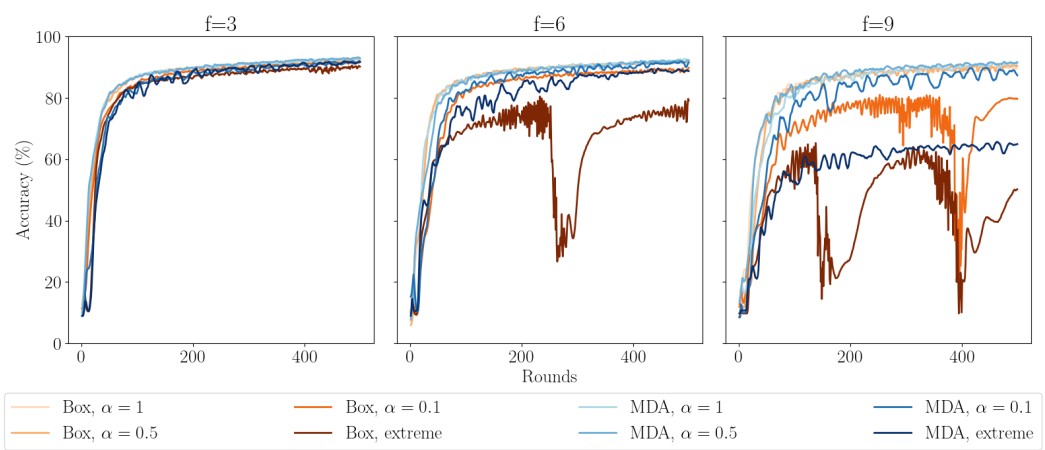

Figure 5: Mimic attack with $f = \{3, 6, 9\}$ in *FedSGD* setting under mild ($\alpha = 1$), moderate ($\alpha = 0.5$), strong ($\alpha = 0.1$) and extreme heterogeneity

2012; Mahloujifar et al., 2019; Farhadkhani et al., 2024). In order to match our theoretical analysis and due to simplicity in implementation, our work considers parameter-based attacks (Shi et al., 2022; Farhadkhani et al., 2022), where the adversary can alter only its messages, while the datasets and the training process remain unchanged.

Our experiments consist of the following attacks:

- **Fall of empires** (FOE): Byzantine clients compute the mean of honest nodes' input, reverse it and scale it by $\epsilon > 0$ Xie et al. (2020). We set $\epsilon = 1$.

- **A little is enough** (ALIE): Byzantine clients estimate mean $\mu$ and standard deviation $\sigma$ of honest nodes and send $\mu - z \cdot \sigma$ to the server Baruch et al. (2019). We set $z = 1$.

- **Sign flip** (SF): inspired by the signSGD algorithm (Jin et al., 2020; Bernstein et al., 2019), the gradient of the faulty clients is multiplied by $-1$ and sent to the server Allen-Zhu et al. (2021). This attack has been widely used in practical simulations (Wu et al., 2020; Wang et al., 2021; Farhadkhani et al., 2022; Xu et al., 2022; Sharma & Marchang, 2024).

- **Mimic**: Byzantine clients imitate one fixed honest client by simply sending its gradient to the server Karimireddy et al. (2022).

## C.2 EXPERIMENTAL RESULTS

***FedSGD* setting:** Figure 5 shows how Box and MDA algorithms perform under the *mimic* attack with $f \in \{3, 6, 9\}$ Byzantine clients. When there are three adversarial clients present in the system, both MDA and Box algorithms converge and achieve up to $93\%$ accuracy. Small accuracy differences appear between mild and extreme heterogeneous distributions, caused by the stronger heterogeneity. With $f = 6$, Box algorithm under all but extreme heterogeneous distribution converge. MDA with extremely heterogeneous datasets seems to converge after expressing instability with $88\%$ accuracy. When the number of Byzantine clients is increased to $f = 9$, Box algorithm under extreme and strong heterogeneity struggles to converge. With mild and moderate heterogeneity, box algorithm converges and reaches over $90\%$ accuracy. MDA is more resilient against the mimic attack, as it converges in mild and moderate heterogeneous setting with over $90\%$ accuracy. In stronger heterogeneity setting, MDA is instable but converges achieving accuracy lower than $65\%$. Since there are 9 adversarial clients ($f = t = 9$) that imitate one honest client (in total 10 clients with the same input), the subset of $n - t$ nodes with the minimum diameter will always contain at least one of these clients. Furthermore, the trusted hyperbox removes $t$ smallest and largest value in each dimension, most likely leaving in the majority of the Byzantine input. Hence, MDA is less influenced by the mimic attack than the Box algorithm.

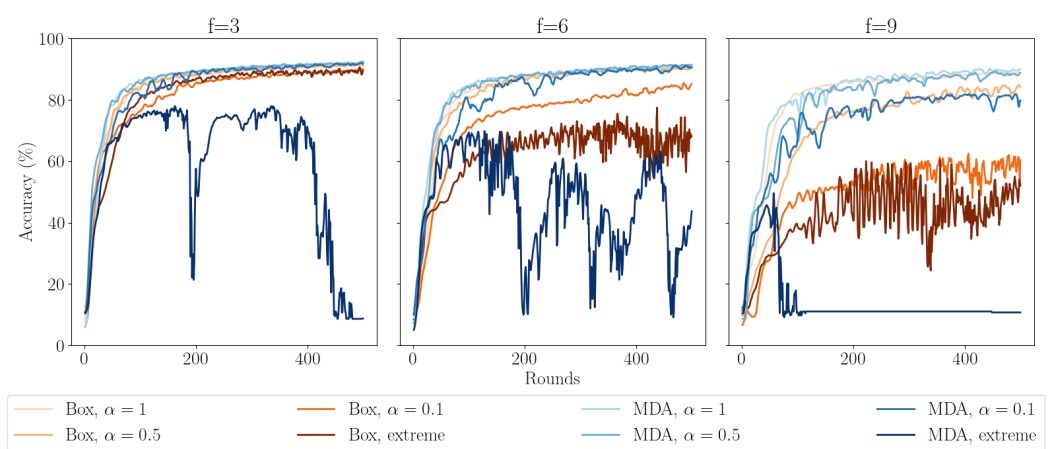

Figure 6: Sign flip attack with $f = \{3, 6, 9\}$ in *FedSGD* setting under mild ($\alpha = 1$), moderate ($\alpha = 0.5$), strong ($\alpha = 0.1$) and extreme heterogeneity

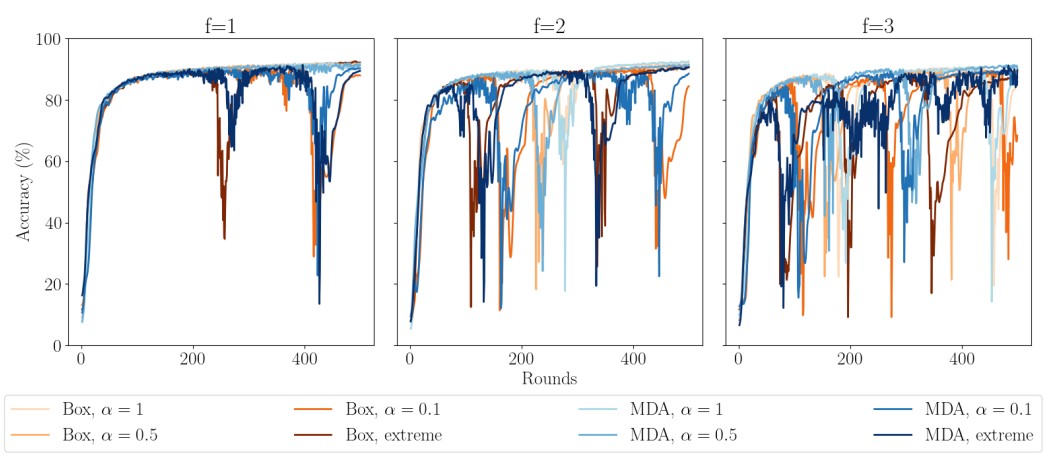

Figure 7: A little is enough attack with $f = \{1, 2, 3\}$ in *FedSGD* setting under mild ($\alpha = 1$), moderate ($\alpha = 0.5$), strong ($\alpha = 0.1$) and extreme heterogeneity

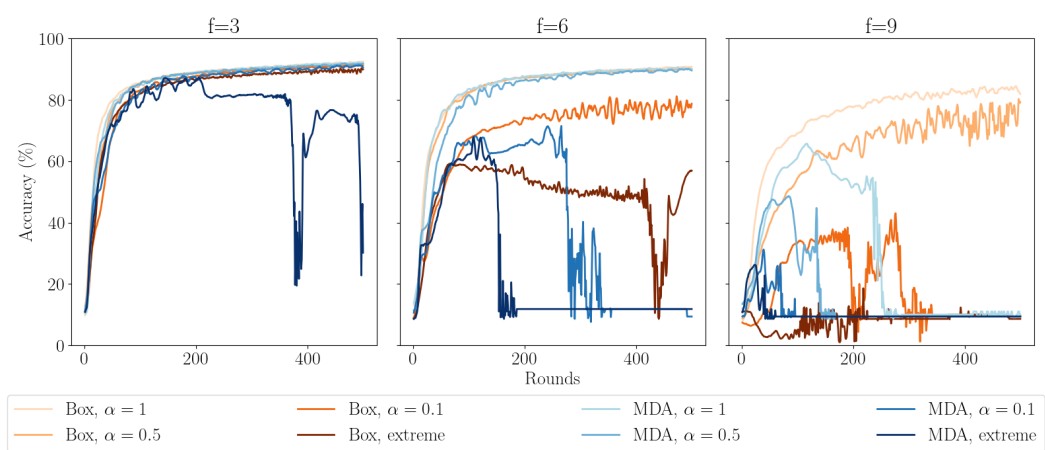

Figure 8: Fall of empires attack with $f = \{3, 6, 9\}$ in *FedSGD* setting under mild ($\alpha = 1$), moderate ($\alpha = 0.5$), strong ($\alpha = 0.1$) and extreme heterogeneity

Figure 6 illustrates the effect of *sign flip* attack on Box and MDA algorithms with $f \in \{3, 6, 9\}$ Byzantine clients. With $f = 3$, MDA and Box algorithms converge reaching $93\%$ and $91\%$ accuracy. In the extreme heterogeneous setting, MDA fails to converge. If the number of adversarial clients is $f = 6$, it can be observed that the Box algorithm is unstable under the extreme heterogeneous setting. With strong heterogeneity, the Box algorithm converges with lower accuracy than MDA, namely $85\%$ compared to $90\%$. When $f = 9$, Box algorithm with strong and moderate heterogeneity converge achieving $62\%$ and $84\%$ accuracy. On the other hand, MDA converges reaching higher accuracy than the Box algorithm, which reflects to our theoretical results showing that MDA is a 2- and Box $2\sqrt{d}$-approximation of the centroid.

Figure 7 depicts performance of Box and MDA algorithms under the *a little is enough* attack with $f \in \{1, 2, 3\}$ adversarial clients. Firstly, we consider a lower number of Byzantine clients, as the attack already affects the system with $f = 2$ and $f = 3$. When there is one adversarial client in the system, both MDA and Box algorithm with mild and moderate heterogeneity converge and reach $92\%$ accuracy. It can be observed that with stronger heterogeneities, both MDA and Box struggle to converge and sudden drops in accuracy occur at regular intervals (around every 250 rounds). With the increased number of Byzantine clients, all algorithms experience these drops. However, as heterogeneity increases, drops in accuracy become more frequent. For example, when $f = 3$, Box algorithm with extreme heterogeneity exhibits accuracy cliffs around every 100 rounds. Overall, the attack induces regular, heterogeneity-dependent accuracy drops that intensify with larger $f$. Both Box and MDA fail to maintain stable convergence under these settings. Note that the batch size is 32. Small batches (e.g., 32) increase variance in honest gradients, making the *a little is enough* and *fall of empires* attacks harder to detect and defend against Karimireddy et al. (2021).

Figure 8 shows how the MDA and the Box algorithms perform under the *fall of empires* attack Xie et al. (2020) with $f \in \{3, 6, 9\}$ Byzantine clients. For $f = 3$, MDA and Box algorithm achieve up to $93\%$ accuracy. However, under extreme heterogeneity, MDA does not converge, and the Box algorithm reaches up to $90\%$ accuracy. For $f = 6$, MDA fails under extreme and strong heterogeneity, whereas Box still converges under strong heterogeneity with lower accuracy. For $f = 9$ Byzantine clients are present, MDA fails across all distributions. However, the Box algorithm converges under mild and moderate heterogeneity and achieves $82\%$ and $75\%$ accuracy, respectively. These results suggest that there may be a trade-off between the centroid approximation and the different validity conditions, also in practice, which we plan to investigate in more detail in future work.

Figure 9 illustrates the Center of $\text{Ball}_{\text{cov}}(S_{\text{Cent}})$ algorithm in the *FedSGD* setting with no Byzantine behavior. It can be observed that after 40,000 rounds $\text{Ball}_{\text{cov}}(S_{\text{Cent}})$ algorithm reaches over $77\%$. The Center of $\text{Ball}_{\text{cov}}(S_{\text{Cent}})$ algorithm requires significantly more rounds than the MDA or the

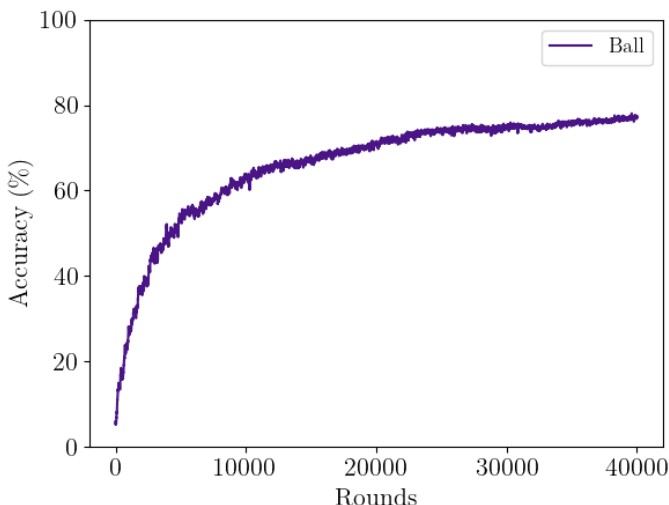

Figure 9: *FedSGD* setting with $\mathrm{Ball}_{\mathrm{cov}}(\mathrm{S}_{\mathrm{Cent}})$ algorithm

Box algorithm and is therefore not evaluated under Byzantine behavior and different heterogeneity distributions.

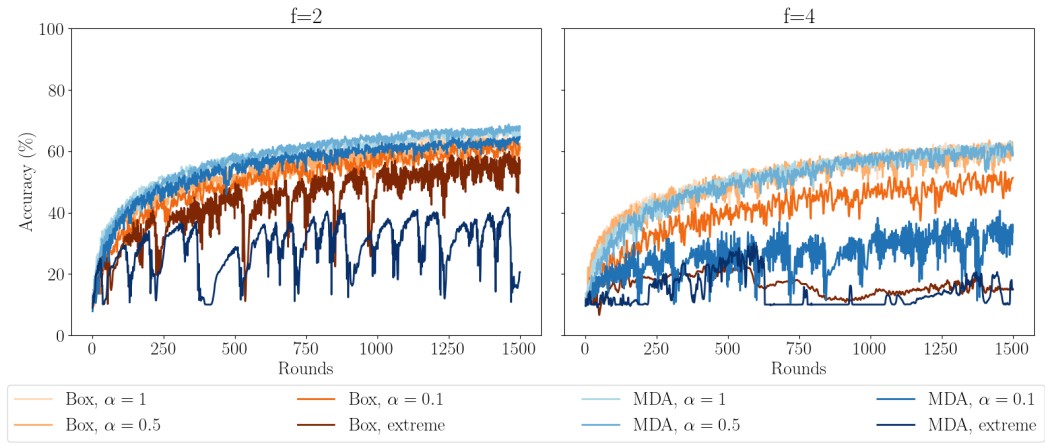

Figure 10: Fall of empires attack with $f = \{2, 4\}$ in *FedSGD* setting under mild ($\alpha = 1$), moderate ($\alpha = 0.5$), strong ($\alpha = 0.1$) and extreme heterogeneity on CifarNet

**CifarNet:** We additionally test the MDA and Box algorithm on CIFAR10 dataset, which has 60,000 $32 \times 32$ color images in 10 classes, out of which 50,000 are training images and 10,000 test images. For the CIFAR10 dataset we implemented CifarNet, a medium-sized convolutional network with thousands of trainable parameters and the ability to capture spatial relationships in colored images. For the experiments, we assume $n = 20$. CIFAR10 is a more complex dataset than MNIST and the experiments require a larger number of training rounds. Hence, we were computationally limited and could test out 20 clients out of which $f = \{2, 4, 6\}$ are faulty.

Figure 10 illustrates the fall of empires attack under 2 and 4 Byzantine clients. Overall, the accuracy of the models training on the CIFAR10 dataset is significantly lower than training on the MNIST dataset. Additionally, with the more heterogeneous setting, fluctuations in accuracy become more evident. Similar to the results in Figure 2, MDA approach achieves a slightly higher accuracy than the Box algorithm in less heterogeneous settings, which complies with the better approximation re-

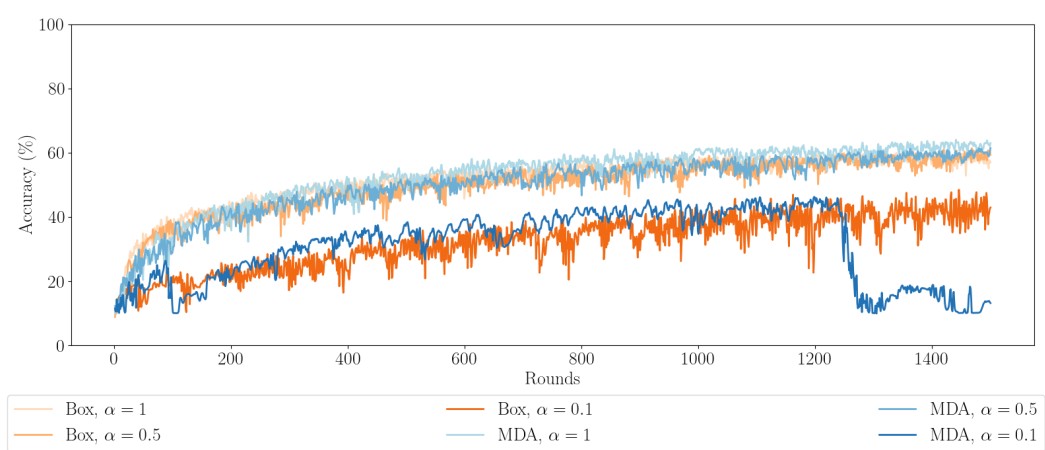

Figure 11: Sign flip attack with $f = 4$ in *FedSGD* setting under mild ($\alpha = 1$), moderate ($\alpha = 0.5$) and strong ($\alpha = 0.1$) heterogeneity on CifarNet

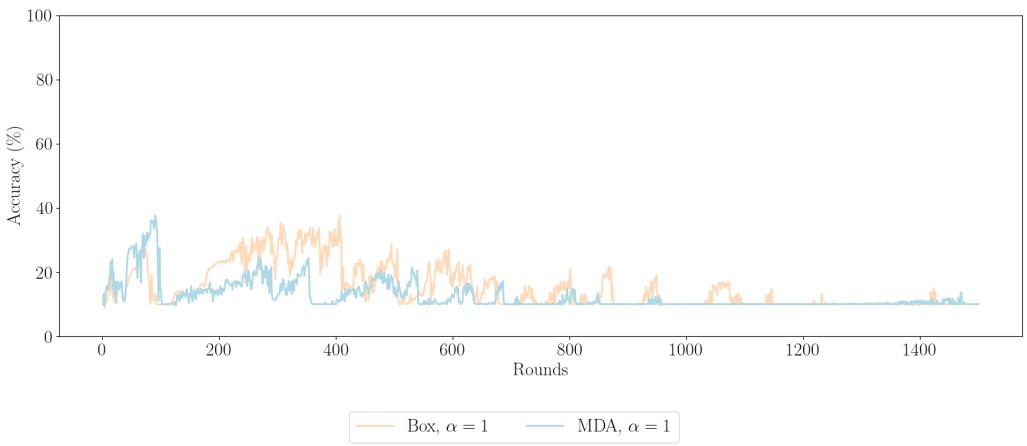

Figure 12: A little is enough attack with $f = 2$ in *FedSGD* setting under mild ($\alpha = 1$) heterogeneity on CifarNet

sults of the centroid. However, with increased heterogeneity between the clients, the Box algorithm outperforms the MDA approach, as it is more robust and satisfies a stronger validity condition than MDA. Hence, we can conclude that stronger validity condition is responsible for providing a more robust solution against Byzantine attacks.

Similar can be concluded from the Figure 11, which depicts the sign flip attack with $f = 4$ under mild, moderate and strong heterogeneous data. Box algorithm converges with low accuracy, whereas MDA fails to converge with strong heterogeneity.

Figure 12 shows a little is enough attack with $f = 2$ with mild heterogeneity. In correspondence to Figure 7 with $f = 3$, MDA and Box fail to converge when $10\%$ of the clients are Byzantine.

Figure 13 illustrates the mimic attack with $f = 6$. Box algorithm does not seem to converge under strong heterogeneity, which complies with the results from Figure 5 with $f = 9$ on MNIST dataset. MDA achieves higher accuracy and is more robust against the mimic attack.

***FedAvg* setting:** Figure 14 illustrates *FedAvg* setting with mild heterogeneous data distribution. In this experiment, we set $f = 1$ and evaluate the algorithms on *sign flip* and *a little is enough*. Additionally, we lower the learning rate to $\eta = 0.001$, since the higher learning rate causes client drift in this setting. In the *sign flip* attack, MDA and Box converge achieving $80\%$ and $78\%$ accuracy,

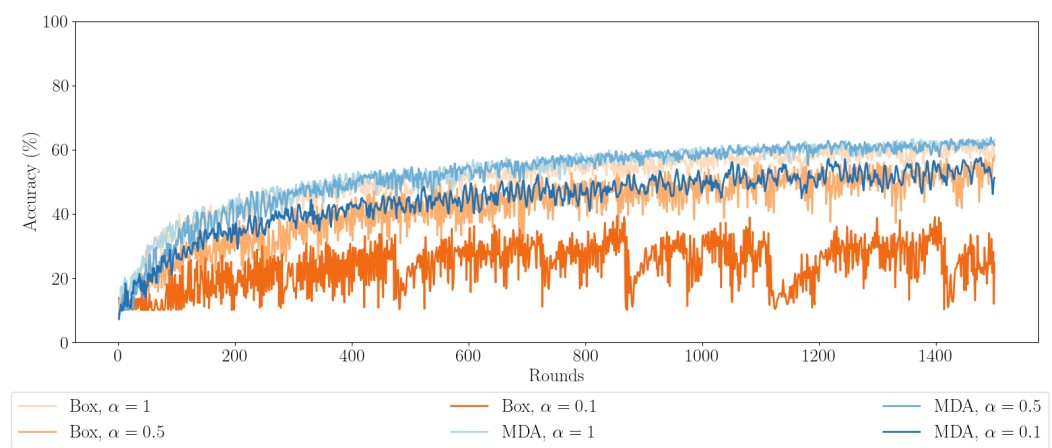

Figure 13: Mimic attack with $f = 6$ in *FedSGD* setting under mild ($\alpha = 1$), moderate ($\alpha = 0.5$) and strong ($\alpha = 0.1$) heterogeneity on CifarNet

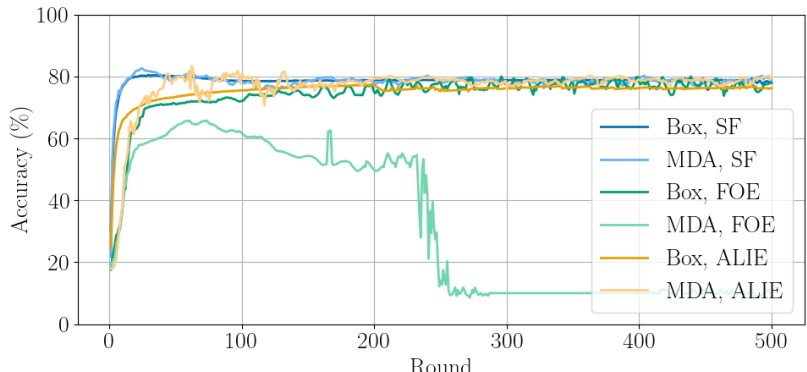

Figure 14: *FedAvg* setting with mild heterogeneous data distributions with $f = 1$

respectively. Nevertheless, MDA shows small differences of $3\%$ in accuracy, whereas the Box algorithm converges smoothly.

In the *a little is enough* attack, accuracy drops slightly to $76\%$ when using the Box algorithm. MDA reaches higher accuracy than Box algorithm ($78\%$), but it is more unstable and shows small differences in accuracy, similar to the ones in the *sign flip* attack.

Under the *fall of empires* attack, Box algorithm converges and reaches $78\%$ accuracy. However, MDA algorithm fails to converge. Compared to the MDA algorithm under *sign flip* or *a little is enough* attack, the *fall of empires* attack has a larger impact and prevents MDA from converging, similar to the results in the *FedSGD* setting.

In future, we intend to continue the empirical evaluation and test out *FedAvg* in different scenarios.

**LLM usage:** The authors used LLMs solely for language editing and clarity improvements. LLMs did not generate ideas, results, proofs, or analyses.

