# OpenReview forum: "Centroid Approximation for Byzantine-Tolerant Federated Learning"
_ICLR.cc/2026/Conference — Submitted to ICLR 2026_

### Official Review · Reviewer_cske · 2025-10-28

**Soundness:** 3
**Presentation:** 3
**Contribution:** 3
**Rating:** 4
**Confidence:** 4

**Summary:**

The authors provide upper bounds (and some lower bounds) for different validity conditions in Byzantine federated learning, offering useful insights into the theoretical guarantees of aggregation algorithms. However, the experimental evaluation does not sufficiently support the theoretical findings.

**Strengths:**

1. The paper is well structured and presented in a clear and coherent manner.
2. The theoretical foundations and analyses are thorough and well supported.

**Weaknesses:**

1. BOX and MDA correspond to the box and centroid validity conditions, respectively. The authors should consider including an aggregation algorithm based on the convex hull of all non-faulty input vectors for comparison. Intuitively, the upper bound under the box condition should be tighter than that under the convex hull, which may imply that the BOX aggregation is more stable.

2. Can the authors provide a theoretical explanation for why BOX converges faster than MDA in the experiments?

3. It appears that the experiments are conducted mainly under the case n < (d + 1)t. The current results are insufficient to align with the theoretical analysis. The authors are encouraged to analyze both cases, n > (d + 1)t and n < (d + 1)t, and provide preliminary empirical verification of the derived upper and lower bounds under different validity conditions (aggregation algorithms).

4. In line 282, when mentioning n < d, there is no such case shown in Table 1. Should it be n < (d + 1)t instead?

**Questions:**

See weakness.

---

> ### Author Response · Authors · 2025-11-21
>
> We would like to thank the reviewer for taking the time to review and provide feedback on our paper. We believe that the revised paper has been significantly improved. Below follows a detailed description of how we addressed each specific comment provided by the reviewer.
>
> ---
>
> > 1. BOX and MDA correspond to the box and centroid validity conditions, respectively. The authors should consider including an aggregation algorithm based on the convex hull of all non-faulty input vectors for comparison. Intuitively, the upper bound under the box condition should be tighter than that under the convex hull, which may imply that the BOX aggregation is more stable.
>
>
> We would like to mention that we do not define a centroid validity condition in this paper. Please note that MDA does not output a vector inside the smallest enclosing ball of the centroid candidate set, as it only achieves a $2$-approximation.
>
> We acknowledge the reviewer’s point regarding convex validity and appreciate the suggestion to evaluate it for comparison. However, we are constrained by the significant computational resources and large datasets required for these experiments, which limit our ability to expand further. To satisfy the convex validity condition, we need to assume that $n > (d+1)t$. That is, just for one Byzantine client, the dimension of the gradients must be smaller than the number of clients. So far, we could expand our experiments to up to 100 clients, whereas the dimension of our smallest dataset (MNIST) is 200k.
>
> From the theoretical perspective, we expect stronger validity conditions to guarantee more stable aggregation. That is, the SafeArea algorithm (Lemma 3.4)  should be more stable than BOX, which is more stable than MDA. On the other hand, the accuracy should be highest for MDA, followed by the BOX and the SafeArea algorithms (because the approximation ratio in one iteration is best for MDA).
>
> To test this setup empirically, we compare the trade-off in validity and approximation for BOX and MDA that seem to support our expectation in the performed experimental evaluation. We hope the reviewer can appreciate this trade-off, as we already face limitations when dividing the MNIST dataset heterogeneously among 100 clients. The computational demands of CIFAR10, on the other hand, are so high that we cannot evaluate it with that many clients.
>
> The corresponding new experimental results can be found in Sections 4 and C2 highlighted in blue in the paper.
>
>
> ---
>
> > 2. Can the authors provide a theoretical explanation for why BOX converges faster than MDA in the experiments?
>
> We believe that this is due to the higher stability (that corresponds to stronger validity) of the BOX algorithm compared to MDA (see previous answer). We provide a new discussion on the stability and stronger validity conditions in Section 4.
>
> ---
>
> > 3. It appears that the experiments are conducted mainly under the case n < (d + 1)t. The current results are insufficient to align with the theoretical analysis. The authors are encouraged to analyze both cases, n > (d + 1)t and n < (d + 1)t, and provide preliminary empirical verification of the derived upper and lower bounds under different validity conditions (aggregation algorithms).
>
> We acknowledge the reviewer's concern that the current results in the evaluation are insufficient to align with the theoretical analysis. We addressed the comments of the reviewers by extending our empirical evaluation to 100 clients for the MNIST dataset and 20 clients for the CIFAR10 dataset. As mentioned in the first answer, computational limitations do not allow us to increase the experimental setup to 200k clients or more. We hope that the reviewer appreciates the theoretical contribution that our paper makes for convex validity and our efforts to get closer to testing convex validity in the future.
>
> ---
>
> > 4. In line 282, when mentioning n < d, there is no such case shown in Table 1. Should it be n < (d + 1)t instead?
>
> The case $n<d$ is included in the results of $n<(d+1)t$ for $t≥1$. For $t=0$, all considered algorithms would be able to compute an optimal solution.
>
> ---
>
> We hope that the above answers properly address the reviewer's concerns. The updated paper has a stronger empirical evaluation and provides more details on the trade-off between the quality of the aggregation rules and the validity conditions. We would like to emphasize that the main contribution of the paper is the robustness definition that has been first adapted for Federated Learning that naturally incorporates Byzantine parties, as well as the corresponding theoretical analysis of Byzantine-tolerant aggregation algorithms. This includes the analysis of the convex validity for which we are not able to evaluate in practice at this point. Considering this important contribution to Byzantine-tolerant federated learning and the extended experimental results in the updated paper, we hope that the reviewer will raise the score of our paper.

---

### Official Review · Reviewer_xg6G · 2025-10-30

**Soundness:** 1
**Presentation:** 1
**Contribution:** 2
**Rating:** 2
**Confidence:** 3

**Summary:**

In this paper the authors present analytical results for centroid approximation in federated learning scenarios with the presence of Byzantine clients. In particular, they show that known validity state-of-the-art conditions do not guarantee good approximation of the average. In particular, through a geometric setting they tighten the upper bound and introduce an improved lower bound.

**Strengths:**

With respect to the majority of FL literature the introduction is fairly original and effectively highlights what are the issues in the state-of-the-art, posing this work under an interesting geometrical perspective.

**Weaknesses:**

After a thorough reading of the paper I am concerned of several aspects, mostly it results unclear and non-immediate what is the true contribution of the work. Therefore, I encourage the authors to be more assertive in their statements and better highlight the novelty of their results, as, in its current state, the work is a little hard to follow.

W1. The mathematical rigour of the definitions should be improved. On one hand it is important to verbally explain definitions and claims, on the other it is important to use a proper notation with symbols, in order to avoid to fall into ambiguity (Definitions 2.3 -- 2.6). Also I think that introducing Notation 1 after its actual use in Def. 2.6.

W2. The overall exposition should be more formal and clear, particularly statements and lemmas.

W3. In the introduction, the authors remark that their results are valid up to a number of attackers $t < n/3$. This sounds a little limiting, as most of the FL literature assumes that $t < n/2$, and now more recent works are moving towards $t< n$.

W4. In the 'Centroid Approximation' setting (lines 179-180), the authors assume that the number of participating clients at each round $m$ is between $n-t$ and $n$. The first concern is that, hence in the worst case scenario, when $ t = n/3$ (or something slightly smaller), the server samples more $2n/3$ clients, causing a non negligible bottleneck -- thus violating the low communication overhead principle in FL. Consider that most of the FL literatures considers federations larger than 100 clients, and the sampled clients are around 10%, or less.Furthermore, as far as I have understood, in order to make the method work, the server should know exactly the number of faulty clients $t$, which is not possible.

W5. The result section is a little bit poor. What happens when you increase the size of the federation $n$? At least another dataset should be evaluated, and more baselines considered.

**Questions:**

See weaknesses.

---

> ### Author Response · Authors · 2025-11-21
>
> We would like to thank the reviewer for taking the time to read through our paper and provide feedback. We believe that the updated paper has been notably improved over the first revisition. In the following, we provide detailed answers to the reviewer´s comments.
>
> ---
>
> > W1. The mathematical rigour of the definitions should be improved. On one hand it is important to verbally explain definitions and claims, on the other it is important to use a proper notation with symbols, in order to avoid to fall into ambiguity (Definitions 2.3 -- 2.6). Also I think that introducing Notation 1 after its actual use in Def. 2.6.
>
> For Definitions 2.3–2.6, we closely follow the notation in the related work, where the approximation ratio under Byzantine adversaries was first introduced. To address the reviewer´s comment, we made a careful pass over the definitions and made corrections that should address the ambiguity. Please refer to the blue text passages in Section 2 “Model and Definitions” to see the changes. As the reviewer correctly pointed out, we have moved Notation 1 before Definition 2.6 and use the notation directly in the definition. With these changes, we believe the updated paper is mathematically rigorous and clear.
>
> ---
>
> > W2. The overall exposition should be more formal and clear, particularly statements and lemmas.
>
> We made another pass over the lemma statements and additionally addressed some corner cases. Please refer to the changes printed in blue in Section 3 of the paper. We believe this made a notable improvement to the paper, and that the formality and clarity of the paper is very good in its current state.
>
> ---
>
> > W3. In the introduction, the authors remark that their results are valid up to a number of attackers $t<n/3$. This sounds a little limiting, as most of the FL literature assumes that $t<n/2$, and now more recent works are moving towards $t<n$.
>
> We thank the reviewer for pointing out that our bound on resilience was not tight in the centralized case. We adjusted our statements to hold for $n>2t$ in the centralized case, and $n>3t$ in the decentralized case. Please note that this corresponds to the best theoretically achievable bounds under Byzantine adversaries for the respective settings. If these bounds do not hold, validity conditions beyond weak validity cannot be satisfied. These results belong to some of the seminal works in distributed computing.
>
> ---

---

> > ### Author Response · Authors · 2025-11-21
> >
> > > W4. In the 'Centroid Approximation' setting (lines 179-180), the authors assume that the number of participating clients at each round $m$ is between $n-t$ and $t$. The first concern is that, hence in the worst case scenario, when $t=n/3$ (or something slightly smaller), the server samples more $2n/3$ clients, causing a non negligible bottleneck -- thus violating the low communication overhead principle in FL. Consider that most of the FL literatures considers federations larger than 100 clients, and the sampled clients are around 10%, or less.Furthermore, as far as I have understood, in order to make the method work, the server should know exactly the number of faulty clients $t$, which is not possible.
> >
> > We would like to clarify that the proposed algorithms do not need to know the exact number of Byzantine clients in the system, as $t<n$ refers to the upper bound on the number of Byzantine clients (see lines 170-171 in the updated version of the paper). This bound can be set in the system design and, as we show in the theoretical part, its value influences the quality of the aggregation vector.
> >
> > To address the communication overhead, we would like to emphasize that the main contribution of this paper is on the theoretical side. From the theoretical perspective, to be able to tolerate up to $\frac{n}{2} -1$ Byzantine clients (aka arbitrary worst-case node failures in a system, see lines 171-174 in the updated version of the paper) in an aggregation step, that is, to prevent an algorithm from choosing an arbitrary worst-case gradient (or model parameters), at least $n$ clients need to participate in an aggregation step. Please note that this type of communication has been studied for federated and collaborative learning in the past, including `[1,2,3]`.
> >
> >
> > To address the comment of the reviewer on the experimental part, we extended the experiments on the MNIST dataset to 100 clients and up to 49 Byzantine failures. This setting is presented in Section 4. As we discuss in Section C1, lines 960-963, this is the maximum number of clients that we can test with highly heterogeneous data. We additionally tested our algorithms on the CIFAR10 dataset with 20 clients. The corresponding results are presented in Section C2.
> >
> > We hope that the reviewer understands that our current experiments are given as a proof of concept for the presented theoretical model with the strongest possible (Byzantine) attacker on the client side.
> >
> > We believe that a model where either Byzantine power or the communication pattern is restricted, e.g., by sampling clients at around 10%, is an interesting direction to extend our approximation measure in future work.
> >
> > ```
> > [1] Adaptive Gradient Clipping for Robust Federated Learning. Allouah et al. ICLR, 2025.
> > [2] Collaborative learning in the jungle (decentralized, byzantine, heterogeneous, asynchronous and nonconvex learning). El-Mhamdi et al. NIPS, 2021.
> > [3] Fixing by mixing: A recipe for optimal byzantine ml under heterogeneity. Allouah et al.  PMLR, 2023.
> > ```
> >
> > ---
> >
> > > W5. The result section is a little bit poor. What happens when you increase the size of the federation $n$? At least another dataset should be evaluated, and more baselines considered.
> >
> > As noted in the previous answer, the updated paper has been extended with more experimental results. See sections 4 and C2 in the updated paper.
> >
> > ---
> >
> > We would once again like to thank the reviewer for the comments, which we believe have been addressed completely. In the revised version of our paper, we have improved the clarity of the writing and increased the number of tolerated Byzantine clients to $n>2t$, which is optimal in the centralized setting. We also extended our experiments to more clients to show that our aggregation rules can be applied in more complex settings. Therefore, we think the score of the paper deserves to be increased in the updated review. We hope the reviewer agrees with this.

---

> > > ### Comment · Reviewer_xg6G · 2025-11-24
> > > **Reply to Authors' Rebuttal**
> > >
> > > I thank the authors for the clear reply. As far as my doubts are concerned, most of them have been properly addressed in the the rebuttal, and the clarity of the updated manuscript has improved. I appreciated the authors' improvement within the experimental section. Therefore, I am consequently raising my score.

---

> > > > ### Author Response · Authors · 2025-11-25
> > > >
> > > > We are pleased to hear that your concerns have been addressed. Thank you for your constructive feedback and for acknowledging our efforts!

---

### Official Review · Reviewer_BerQ · 2025-11-01

**Soundness:** 3
**Presentation:** 3
**Contribution:** 2
**Rating:** 2
**Confidence:** 3

**Summary:**

The paper analyzes Byzantine-tolerant federated learning through the lens of centroid approximation. It defines the set of all (n−t)-subset centroids, the minimum covering ball around those centroids, and measures any aggregator by how close its output is to the true non-faulty centroid relative to the ball’s radius. Using validity conditions drawn from distributed agreement, such as weak, strong, box, and convex validity, the authors derive upper and lower bounds on the achievable approximation ratios, with new and mostly tight results for box validity and clear statements for convex validity. They also argue that the guarantees extend to synchronous peer-to-peer FL by combining deterministic aggregation with Byzantine agreement, and they provide preliminary simulations on MNIST under several parameter-based attacks that suggest a practical trade-off between approximation quality and validity constraints.

**Strengths:**

1. The definitions of candidate centroids, minimum covering ball, and the centroid approximation ratio are crisp and useful for analyzing Byzantine aggregation beyond worst-case distances.
2. Upper/lower bounds under weak/strong/box/convex validity are tabulated; the new analysis for box validity (including n<d) and the tight 2d bound for convex validity are valuable.
3. The extension argument to peer-to-peer (interactive consistency → identical inputs → deterministic aggregation) is straightforward and helps position the theory for decentralized FL.

**Weaknesses:**

1. The setup treats all clients equally (ignores sample-size heterogeneity) “to restrict Byzantine power,” which deviates from standard sample-count weighting in FL. Please justify this modeling choice and discuss implications for deployment, including whether your bounds/algorithms still hold under non-uniform weights or can be adapted with importance weighting.
2. (Eq. 3) is stated for general n,d,t but never checks corner cases like t=0, n≤2t, n<d, or n=d+1. The statement should list validity regimes explicitly and clarify vacuous/impossible cases.
3. (Eq. 5) relies on a denominator that can be zero for certain client selections (e.g., when all trimmed coordinates coincide); the text doesn’t specify the fallback (skip, clip, or add ε).
4. The step from (Eq. 7) to (Eq. 8) uses an ℓ2bound and then applies an ℓ∞ argument without a stated inequality tying them together.
5. The inequality used from (Eq. 9) → (Eq. 10) assumes convexity/linearity in the wrong place; it moves the norm outside an average without justifying it.
6. (Eq. 11) bounds a gradient/model drift term but never states the Lipschitz or smoothness constants required to make that bound legal.
7. (Eq. 12) exchanges 𝐸 with a coordinate-wise trimming/selection; measurability and selection dependence on the sample aren’t addressed.

**Questions:**

See Weaknesses.

---

> ### Author Response · Authors · 2025-11-21
>
> We would like to thank the reviewer for taking the time to read our paper and provide feedback on our work. We have done our best to address all comments, and believe that the updated paper has been significantly improved. In the following, we provide answers to the specific questions raised by the reviewer.
>
> ---
>
> > 1. The setup treats all clients equally (ignores sample-size heterogeneity) “to restrict Byzantine power,” which deviates from standard sample-count weighting in FL. Please justify this modeling choice and discuss implications for deployment, including whether your bounds/algorithms still hold under non-uniform weights or can be adapted with importance weighting.
>
> We understand that the phrasing we used in the paper about restricting Byzantine power is ambiguous. What we meant was the following: If we assume that some clients have more data and let the algorithm treat the clients differently by putting more weight on gradients coming from larger datasets, we would have to provide bounds on the number of Byzantine clients in terms of batch sizes. Such an approach either restricts the Byzantine power, if one assumes that the server hands out the datasets, or it does not allow an algorithm to tolerate as many Byzantine clients, if the clients are allowed to report their batch sizes (and Byzantine clients can thus lie about the data size).
>
> Instead, when deriving theoretical bounds and presenting algorithms, the more common approach in the literature is to assume that all clients are treated equally, as if they had datasets of the same size, and base the discussion on the number of clients that an algorithm can tolerate.
>
> We would also like to draw the reviewer’s attention to the evaluation section, where we test our algorithms on heterogeneous data distributions, where the clients have datasets of different sizes.
>
> To address this ambiguity in our work, we adjusted the phrasing of the corresponding sentence in the model section (please refer to the new text passage in blue in lines 174-176).
>
> ---
>
> > 2. (Eq. 3) is stated for general n,d,t but never checks corner cases like t=0, n≤2t, n<d, or n=d+1. The statement should list validity regimes explicitly and clarify vacuous/impossible cases.
>
> We thank the reviewer for the comment, and we clarify the corner cases in the following:
>
> $t=0$: the centroid of all n vectors is Cent*, and the centroid box is also just one centroid. The lower bound in Lemma 3.3 does not hold in this case, as Cent* can be commuted in this case. Please note that we added this to the lemma statement (please refer to the new text passage in blue in line 252-253).
>
> $n≤2t$: this case is not covered by the lemma, as we assume $n>2t$ in our work.
>
> $n<d$: in this case $(n-t)/t <d$, and thus the lower bound is $(n-t)/t$. For this case, we mention in the paper (see line 288) that the bounds are not tight.
>
> $n=d+1$: For $t>0$, the lower bound is $\sqrt{(n-t)/2t}$, as $(n-t)/t≥d$
>
> ---
>
> > 3. (Eq. 5) relies on a denominator that can be zero for certain client selections (e.g., when all trimmed coordinates coincide); the text doesn’t specify the fallback (skip, clip, or add ε).
>
> By assuming that $t>0$ in the lemma statement and $x>0$ in the lower bound description, the radius of the ball cannot be $0$. To address the reviewer comment, we adjusted the lemma statement and the proof accordingly (please refer to the new text passage in blue in lines 252-253, line 359, and line 376).
>
> ---
>
> > 4. The step from (Eq. 7) to (Eq. 8) uses an ℓ2bound and then applies an ℓ∞ argument without a stated inequality tying them together.
>
> We think that this comment refers to Lemma 3.4, but we cannot be sure as we do not number the equations in our paper. We do not use norm arguments in the proof of Lemma 3.4, since we bound the terms directly using averages of the smallest and the largest t values in each coordinate. We would be happy to provide more details about the analysis if the reviewer would kindly point us to exactly which part of the paper (page and line number) that this comment refers to.
>
> ---
>
> > 5. The inequality used from (Eq. 9) → (Eq. 10) assumes convexity/linearity in the wrong place; it moves the norm outside an average without justifying it.
>
> We are unsure which part of the paper the reviewer refers to and would kindly like to ask the reviewer to point us to the corresponding line numbers.
>
> ---
>
> > 6. (Eq. 11) bounds a gradient/model drift term but never states the Lipschitz or smoothness constants required to make that bound legal.
>
> We are unsure which part of the paper the reviewer refers to and would kindly like to ask the reviewer to point us to the corresponding line numbers.
>
> ---

---

> > ### Author Response · Authors · 2025-11-21
> >
> > > 7.(Eq. 12) exchanges 𝐸 with a coordinate-wise trimming/selection; measurability and selection dependence on the sample aren’t addressed.
> >
> > We are unsure which part of the paper the reviewer refers to and would kindly like to ask the reviewer to point us to the corresponding line numbers.
> >
> > ---
> >
> >
> > With the exception of a few comments where we need more input from the reviewer, we hope that the reviewer is satisfied with our answers, which we believe improved the clarity and presentation of the paper. Therefore, we hope that the reviewer is open to increasing the score of the paper.

---

### Official Review · Reviewer_1PeB · 2025-11-04

**Soundness:** 2
**Presentation:** 2
**Contribution:** 2
**Rating:** 4
**Confidence:** 1

**Summary:**

This paper studies the theoretical and empirical foundations of aggregation robustness in Byzantine-tolerant federated learning (FL). The authors analyze the centroid approximation problem, i.e., how closely an aggregation algorithm can approximate the average of non-faulty clients when up to t out of n clients are Byzantine. They establish lower and upper bounds for centroid approximation under different validity conditions, provide almost-tight bounds for box validity, and extend the results to peer-to-peer settings. Theoretical analysis is illustrated by experiments under various Byzantine attacks and heterogeneous data distributions.

**Strengths:**

1. The paper provides the first lower bound of $\min \{n/t - 1, \sqrt{d} \}$ for centroid approximation under box validity and an improved upper bound of $2\min \{n , \sqrt{d} \}$ establishing nearly tight limits.

2. Bridging approximate agreement theory and federated learning offers a good perspective and strengthens theoretical rigor in Byzantine robustness. Extension to a peer-to-peer network setting is provided.

**Weaknesses:**

The main concern is in the simulation part. The experimental setting (30 clients) is small. It remains unclear how the proposed bounds or algorithms behave on large-scale FL systems with realistic models (e.g., CNNs, Transformers).

**Questions:**

Could the authors provided more experimental results on more practical settings and other complex tasks?

---

> ### Author Response · Authors · 2025-11-21
>
> We would like to thank the reviewer for taking the time to review our paper. By addressing the comments, we believe the quality of the paper has been notably improved. In the following, we provide a detailed answer to each comment.
>
> ---
>
> > 1. The main concern is in the simulation part. The experimental setting (30 clients) is small. It remains unclear how the proposed bounds or algorithms behave on large-scale FL systems with realistic models (e.g., CNNs, Transformers).
>
> We agree with the reviewer that it is important to collect experimental results in a relevant setting. We therefore performed more experiments. The results are included in the revised version of the paper. In Section 4, we replaced the experimental setting with 30 clients on the MNIST dataset by experiments with 100 clients, and up to 49 Byzantine faults. We would like to emphasize that with 100 clients we are reaching the limit of clients that we can consider while distributing the data heterogeneously among the clients, which is explained in Section C1.
>
> We additionally tested our algorithms on the CIFAR10 dataset with 20 clients. The corresponding results are presented in Section C2 (Figures 10-13). Unfortunately, we do not currently have the computational resources to expand our experiments to much larger datasets. This, together with evaluations on other deep learning architectures, is planned for future work. Despite this, we believe the experiments in our paper are an important contribution to literature on Federated Learning since they underline the importance of considering robustness as a trade-off between validity conditions and the quality of the aggregation when Byzantine clients are present in a system.
>
> ---
>
> > 2. Could the authors provided more experimental results on more practical settings and other complex tasks?
>
> We agree with the reviewer that extending Byzantine-tolerant FL evaluations to more practical settings is an important direction for future work. As noted earlier, we have put great effort into extending our experimental results in the updated paper, shown in Section 4 and Section C2. We hope that the reviewer appreciates these efforts. Although the datasets considered are still small compared to conventional FL datasets, we face challenges in tackling them under Byzantine clients, as has also been pointed out in related research (see `[1,2,3,4]`). We also want to emphasize that our current experiments are given as a proof of concept, and that we are constrained by the significant computational resources required for these experiments.
>
> ```
> [1] Adaptive Gradient Clipping for Robust Federated Learning. Allouah et al. ICLR, 2025.
> [2] Learning from history for byzantine robust optimization. Karimireddy et al. PMLR, 2021.
> [3] Approximate Agreement Algorithms for Byzantine Collaborative Learning. Cambus et al. SPAA, 2025.
> [4] Byzantine machine learning made easy by resilient averaging of momentums. Farhadkhani et al. PMLR, 2022.
> ```
>
> ---
>
> We hope that the above answers properly address the reviewer's concerns. We would like to emphasize that our contribution lies in defining robustness of aggregation rules as a trade-off between the quality of a centroid and the robustness given by the validity conditions, thus making a first step toward designing truly Byzantine-tolerant federated learning algorithms. We hope that this theoretical contribution and the new experimental results on more complex tasks will convince the reviewer that the updated paper deserves a higher score.

---

### Official Review · Reviewer_Lv4r · 2025-11-04

**Soundness:** 3
**Presentation:** 3
**Contribution:** 2
**Rating:** 4
**Confidence:** 3

**Summary:**

The paper introduces a novel theoretical framework for Byzantine-tolerant Federated Learning (FL) leveraging the concept of centroid approximation. In contrast to other methods that typically aim to detect or exclude faulty clients, the paper proposes to keep all updates and evaluate how well an aggregation rule can approximate the true average of the non-faulty clients, even when some updates are Byzantine. The paper uses ideas from distributed computing and FL, analyzing how different validity conditions constrain the quality of possible approximations. The theoretical contributions include 1) a lower bound of the centroid approximation under box validity, 2) an upper bound by providing a new analysis for the case when the number of clients is smaller than the number of parameters, and 3) an algorithm that achieves a sqrt(2d) approximation under convex validity. The paper also includes experiments with SGD and federated averaging aggregation schemes.

**Strengths:**

+ The theoretical analysis is sound and establishes nearly tight upper and lower bounds on centroid approximation under different validity conditions, extending classical results from distributed computing to FL.
+ The angle of using the centroid approximation metric as a way to analyze Byzantine robustness is interesting. The paper clearly explains the geometric intuition. On the other hand, the idea of not excluding the Byzantine clients (unlike other aggregation methods) and yet achieving robust aggregation is also quite interesting.
+ The proofs are well structured and grounded and the theoretical contributions includes both lower and upper bounds.
+ The authors show that the results can be easily extended to peer-to-peer FL scenarios.
+ The experiments help to support the theoretical findings and contributions.

**Weaknesses:**

+ The safe area assumption (t < n/(d+1)) in Definition 2.8 is quite impractical for most FL scenarios, collapsing to t=0 in realistic settings where d is large. Then, some of the results, like Lemma 3.4, although mathematically valid, has a limited relevance for low-dimensional problems.
+ Some of the assumptions are also restrictive for many practical FL scenarios: the paper assumes synchronous communication, equal local training data sizes, or static participation. It helps to simplify the theoretical analysis but is realistic for many practical FL deployments.
+ Although the centroid approximation provides an interesting view on robustness, its link to actual learning performance (like convergence or accuracy) remains unquantified in the paper. In other words, how does centroid approximation connect with convergence during training or with accuracy?
+ The experimental evaluation uses simple settings: all experiments use a single dataset (MNIST), a simple MLP model, and basic FedSGD and FedAvg setups. On the other hand, while several attacks are tested (in the appendix), the empirical study lacks diversity in data, model architectures, or comparison to other robust aggregation methods in the state-of-the-art.

**Questions:**

+ Given that (t < n/(d+1)) becomes infeasible when d is much bigger than n, do you see any realistic settings where convex-validity results (like in Lemma 3.4) could apply? Is it possible to relax this assumption to make this theoretical framework more usable in practice?
+ Although the proposed sqrt(2d) approximation algorithm seems theoretically sound, could you explain how does this algorithm scale for models with millions of parameters or in scenarios with a large number of clients?
+ Related to the previous question on computational complexity: can the algorithm be applied to other architectures beyond an MLP (e.g., CNNs, transformers)?
+ Have you empirically checked whether smaller centroid approximation ratios correlate with faster convergence or higher accuracy?
+ Have you considered adaptive attacks for evaluating the robustness of the algorithm?

---

> ### Author Response · Authors · 2025-11-21
>
> We would like to thank the reviewer for the valuable feedback. By addressing the questions, we believe that the quality of the paper has improved substantially. In the following, we provide detailed answers to the questions raised in the review.
>
> ---
>
> >1. Given that (t < n/(d+1)) becomes infeasible when d is much bigger than n, do you see any realistic settings where convex-validity results (like in Lemma 3.4) could apply? Is it possible to relax this assumption to make this theoretical framework more usable in practice?
>
> The convex validity condition is extensively used in the literature to solve multidimensional Byzantine agreement in distributed computing. It is also a natural validity condition that, by definition, removes all Byzantine vectors that could potentially harm an algorithm (at a cost of a worse approximation ratio). This is the main reason we consider the convex validity condition in this work.
>
> With regard to applicability, we believe that there are situations where this validity condition is applicable. Consider for example the case where a large number of mobile devices are used to train a relatively small machine learning model. Unfortunately, we do not currently have access to the computational resources to study this case in practice. Therefore, our results on convex validity should primarily be considered a theoretical contribution to the community.
>
> Unfortunately, it does not seem to be possible to relax the convex validity assumption. The authors in `[1]` showed that also relaxations of the convex validity fail to break the $t < n/(d+1)$ bound. They also show that box validity is the only relaxation based on projections that can achieve the $n>3t$ bound in the distributed setting. We added a comment about this in the contribution section of the paper (see the new text passage in blue on lines 99 and 103 in the revised version).
>
> ```
> [1] Relaxed Byzantine Vector Consensus.Zhuolun Xiang and Nitin H. Vaidya. Leibniz International Proceedings in Informatics (LIPIcs), Volume 70, pp. 26:1-26:15, Schloss Dagstuhl – Leibniz-Zentrum für Informatik, 2017.
> ```
>
> ---
>
> > 2. Although the proposed sqrt(2d) approximation algorithm seems theoretically sound, could you explain how does this algorithm scale for models with millions of parameters or in scenarios with a large number of clients?
>
> The algorithm is efficient in terms complexity of computing the aggregation vector. This computation is dominated by the trimming step, where the largest and smallest $t$ values are removed in each coordinate. Then, depending on the algorithm used to compute the approximation, either the average of the remaining vectors or an intersection with the box containing all possible centroids is calculated. This can be done in $O(n\log(t))$ sequential time, or even faster with parallelization. The complexity of computing the aggregation vector is not a bottleneck in our experiments.
>
> The quality of the aggregation vector depends either on the number of clients or the dimension of the model, as it is computed as $2\sqrt{\min(n,d)}$. In our experiments, we have $n\ll d$, and thus the aggregation quality scales with $\sqrt{n}$.
>
> ---
>
> > 3. Related to the previous question on computational complexity: can the algorithm be applied to other architectures beyond an MLP (e.g., CNNs, transformers)?
>
> Yes, our algorithms can be applied to other architectures as the aggregation step is independent of the architecture. To address this, in Section C2, we provide new experiments for the CIFAR10 dataset that use CifarNet as the architecture.
>
> ---
>
> > 4. Have you empirically checked whether smaller centroid approximation ratios correlate with faster convergence or higher accuracy?
>
> Our experiments suggest that there is a correlation between the validity conditions and the robustness of the solution, and between smaller centroid approximation and accuracy. We improved the discussion in Section 4 to also include this trade-off.
>
> However, it is still not clear how much the centroid approximation and the validity conditions influence the performance of the trained model. We believe that our practical results rather underline our theoretical contribution which shows that there is a large gap between aggregation algorithms that provide the best possible approximation while only satisfying the weak validity condition (or weaker conditions) versus algorithms that provide strong validity, but have a bad approximation. In future work, we plan to investigate this trade-off and find even better algorithms for Byzantine-tolerant FL.
>
> ---

---

> > ### Author Response · Authors · 2025-11-21
> >
> > > 5. Have you considered adaptive attacks for evaluating the robustness of the algorithm?
> >
> > So far, we have considered the attacks “Fall of empires”, “A little is enough”, “Mimic”, and “Sign flip”. Our implementations of FOE, ALIE, and Mimic use round-wise estimates of the honest clients’ mean and variance, and are thus attacks that are adjusted in every round. However, they are not adaptive attacks in the strict sense because their update rule does not change between rounds in response to previous model states. In future work, we plan to extend our setting and evaluate fully adaptive attacks.
> >
> > ---
> >
> > We hope that the above answers properly address the reviewer's concerns. We believe that the main focus of the paper is the theoretical contribution with the first robustness analysis of aggregation algorithms in Federated Learning that works under Byzantine adversaries. The extended experimental results reflect the trade-offs that we see in the theory and show that the aggregation rules can also be applied to more complicated machine learning models. Considering this important contribution to Byzantine-tolerant federated learning, we hope that the reviewer would be open to increasing the score of the paper.

---

### Author Response · Authors · 2025-12-02

Dear Area Chair,

In the following, we would like to quickly summarize the discussion and rebuttal period for your convenience.

Thanks to the very helpful comments from the reviewers, we significantly revised and extended our paper during the rebuttal period, both on the theoretical and on the empirical front. Below, we briefly summarize the main points, however we have addressed all other reviewer comments as well. Please also find attached the new version of the paper where all changes and additional results are highlighted in blue color.

The first key change in our paper is the tight bound on resilience in the centralized case. Our statements now hold for n>2t clients, which is the best achievable bound under Byzantine adversaries. The second key change includes additional experiments on the MNIST dataset with 100 clients and up to 49 Byzantine clients, found in Section 4 of the improved manuscript. We also provide new experimental results on the CIFAR10 dataset and underlying CifarNet convolutional network with 20 clients, shown in Section C2. Furthermore, we fixed the notation concerns and improved the clarity of the manuscript.

We believe we have addressed all reviewers' comments. Both the theoretical and experimental parts of our paper have improved substantially, which was acknowledged by one reviewer that engaged in the discussion with us and increased the score. We were however hoping for a longer discussion with all reviewers, as we believe we have addressed all their concerns and improved the paper significantly.

---

### Meta-Review · Area_Chair_UjNu · 2026-01-07

**Summary:**

The paper proposes the computation of a centroid (defined in lines 360-363) to solve the challenging issue of adversarially robust distributed learning (adversarial during the training phase), often called Byzantine tolerant ML.

The problem is still very timely despite several published/established solutions, in particular, the authors, bring a framework from classical distributed computing called "approximate agreement". While this is an original angle, it brings with it the inherited limitations of AA and was not able to convince a ML audience of its utility.

In particular, as Reviewer Lv4r's pointed, the concern on the bound involving t (the number of tolerated faulty nodes), n (the total number of nodes) and d (the dimension of the parameter vector) seems to be impossible to tackle using the agreement framework proposed by the authors. It remains imho the reason to look for other solutions outside this framework.

The concerns on the running time of the aggregation seems minor, those on the quality of the aggregation w.r.t. dimension of the model fall under usual limitations of robust statistics (sqrt(d) scaling). Reviewers (such as xg6G) also raised several other concerns that the authors tackled despite the context of this year's ICLR, which is very appreciated.

Notes from the AC :
- in any re-submission of the paper, Figures 2, 3 and 4 in the main, and 5 to 14 in the appendix, need to have at least some error bar or show how the plots repeat over several runs.
- it remains unclear wether AA brings any *practical* addition to realistic ML (when d>>n), the paper needs to work out that aspect more thoroughly in upcoming iterations.

**Reviewer Concerns:**

most minor issue from reviewers are tackled, e.g. those from xg6G), the most important issue was raised by Lv4r and was not addressed (bound on t, n and d: t < n/(d+1) )

**Reviewer Scores:**

xg6G would have raised from 2 to probably 4

---

### Decision · Program_Chairs · 2026-01-26

Reject